# Self-Imitation Learning via Trajectory-Conditioned Policy for Hard-Exploration Tasks

## Abstract

Imitation learning from human-expert demonstrations has been shown to be greatly helpful for challenging reinforcement learning problems with sparse environment rewards. However, it is very difficult to achieve similar success without relying on expert demonstrations. Recent works on self-imitation learning showed that imitating the agent's own past good experience could indirectly drive exploration in some environments, but these methods often lead to sub-optimal and myopic behavior. To address this issue, we argue that exploration in diverse directions by imitating diverse trajectories, instead of focusing on limited good trajectories, is more desirable for the hard-exploration tasks. We propose a new method of learning a trajectory-conditioned policy to imitate diverse trajectories from the agent's own past experiences and show that such self-imitation helps avoid myopic behavior and increases the chance of finding a globally optimal solution for hard-exploration tasks, especially when there are misleading rewards. Our method significantly outperforms existing self-imitation learning and count-based exploration methods on various hard-exploration tasks with local optima. In particular, we report a state-of-the-art score of more than 20,000 points on Montezumas Revenge without using expert demonstrations or resetting to arbitrary states.

## 1 Introduction

Hard-exploration tasks, particularly characterized by sparse environment rewards, are traditionally challenging in reinforcement learning (RL), because the agent must carefully balance the exploration and exploitation when taking a long sequence of actions to receive infrequent non-zero rewards. Demonstration data has been shown to be helpful for tackling hard-exploration problems (Subramanian et al., 2016); many existing methods (Hester et al., 2018; Pohlen et al., 2018; Aytar et al., 2018; Salimans & Chen, 2018) provide the guidance for exploration based on imitation learning of expert demonstrations and achieve strong performances on hard-exploration tasks. However, the reliance on human demonstrations largely limits the general applicability of such approaches.

The agent's own past good trajectories with high total rewards are easily accessible (though imperfect) alternatives for the human-expert demonstrations. Recent works (Oh et al., 2018; Gangwani et al., 2018) verify that imitation learning from the agent's previous good trajectories could indirectly drive exploration in certain environments. However, imitation of good experiences within limited directions might hurt exploration in some cases. Specifically, in environments with misleading rewards which may trap the agent in local optima, simply imitating 'good' trajectories that would accumulate misleading positive rewards may guide the agent to a myopic behavior and hinder it from reaching a higher return in the longer term. Therefore, imitating diverse trajectories would

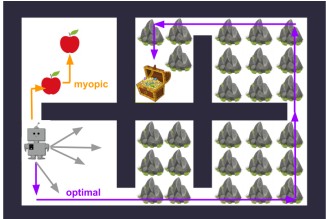

Figure 1: Map of Apple-Gold domain, where the reward for getting an apple, getting the gold and taking a step in the rock is 1, 10, -0.05 respectively. The time limit for one episode is 45 steps.

be more desirable to help encourage exploration in diverse directions and avoid being distracted by the misleading rewards. For example, as illustrated in Figure 1, the agent starts in the bottom left corner where it can easily collect apples near its initial location by random exploration and achieve a small positive reward. If the agent imitates the trajectories around the orange path, it would receive the nearby positive rewards quickly but it is unlikely to collect the gold within a given time limit. Therefore, in order to find the optimal path (purple), it is better to exploit the past experiences in diverse directions (gray paths), instead of focusing only on the trajectories with the myopic behavior.

This paper investigates how imitation of diverse past trajectories leads a further exploration and helps avoid getting stuck at a sub-optimal behavior. Our main contributions are summarized as follows: (1) We propose a novel architecture for a trajectory-conditioned policy that can imitate diverse demonstrations. (2) We show the importance of imitating diverse past experiences to indirectly drive exploration to different regions of the environment, by comparing to existing approaches on various sparse-reward reinforcement learning tasks with discrete and continuous action space. (3) We achieve a performance comparable with the state-of-the-art on hard-exploration Atari game of Montezuma's Revenge and Pitfall without using expert demonstrations or resetting to arbitrary states.

## 2 RELATED WORK

**Imitation Learning**   The goal of imitation learning is to train a policy to mimic a given demonstration. Many previous works achieve good results on hard-exploration Atari games by imitating human demonstrations. DQfD  (Hester et al., 2018) combines the temporal difference updates in Q-learning with the supervised classification of the demonstrators' actions. Ape-X DQfD (Pohlen et al., 2018) extends DQfD with the transformed Bellman operator and temporal consistency loss to improve the efficiency of exploration. Aytar et al. (2018) learn embeddings from a variety of demonstrations videos and proposes the one-shot imitation learning reward, which inspires the design of reward in our method. All these successful attempts rely on the availability of human demonstrations. In contrast, our method treats the agent's past trajectories as demonstrations. Imitation learning is more difficult when the environment becomes more stochastic because the demonstrations could not account for all possible situations. Our method allows for some flexibility to follow the demonstrations in a soft-order and thus could perform well in the environment with a moderate degree of stochasticity. As discussed in Appendix C.1, we can easily extend our method to handle a more challenging scenario (e.g., where the location of objects could be random). Yet, imitation learning in extremely stochastic environments is still an open problem (Ghosh et al., 2017; Paine et al., 2019).

**Self-Imitation**   Learning a good policy by imitating past experiences has been discussed where the agent is trained to imitate only the high-reward trajectories with the SIL (Oh et al., 2018) or GAIL objective (Gangwani et al., 2018). In contrast, we store the past trajectories ending with diverse states in the buffer, because trajectories with low reward in the short term might lead to high reward in the long term, and thus following a diverse set of trajectories could be beneficial for discovering optimal solutions. Furthermore, our method focuses on explicit trajectory-level imitation while existing methods use sampled state-action pairs from the buffer to update the policy. Gangwani et al. (2018) proposed to learn multiple diverse policies in a SIL framework using the Stein Variational Policy Gradient. Empirically, their exploration can be limited by the number of policies learned simultaneously and the exploration performance of every single policy, as shown in Appendix F.

**Exploration**   Many exploration methods (Schmidhuber, 1991; Auer, 2002; Chentanez et al., 2005; Strehl & Littman, 2008) in RL tend to award a bonus to encourage an agent to visit novel states. Recently this idea was scaled up to large state spaces (Tang et al., 2017; Bellemare et al., 2016; Ostrovski et al., 2017; Burda et al., 2018). We propose that instead of directly taking a quantification of novelty as an intrinsic reward, one can encourage exploration by rewarding the agent when it successfully imitates demonstrations that would lead to novel states and gain the advantages in exploitation, as discussed in Appendix I. Go-Explore (Ecoffet et al., 2019) also shows the benefit of exploration by returning to promising states. Our method can be viewed in general as an extension of Go-Explore, though we do not need to explicitly divide learning into two phases of exploration and robustification. Go-Explore relies on the assumption that the environment is resettable. Resetting to an arbitrary state is often infeasible in real environments and gives an unfair advantage. When using a perfect goal-conditioned policy instead of a direct 'reset' function, this variant of Go-Explore may not explore as efficiently as our method, as discussed in Appendix H. Previous works attempted reaching a goal state by learning a set of sub-policies (Liu et al., 2019) or a goal-conditioned policy in pixel observation space (Dong et al., 2019). Gregor et al. (2016); Eysenbach et al. (2018); Pong et al. (2019) seek a diversity of exploration by maximizing the entropy of mixture skill policies or generated goal states. However, these methods do not show experimental results performing well on sparse-reward environments with a rich observation space like Atari games.

**Goal-Conditioned Policy**   Andrychowicz et al. (2017); Nair et al. (2017); Schaul et al. (2015a); Pathak et al. (2018) studied learning a goal-conditioned policy. Our trajectory-conditioned policy could be viewed as a goal-conditioned policy. Similarly to hindsight experience replay (Andrychowicz et al., 2017), our approach samples goal states from past experiences. Compared to conditioning on a single final goal state, the state trajectory includes rich intermediate information leading the

---

**Algorithm 1** Diverse Self-Imitation Learning with Trajectory-Conditioned Policy

---

Initialize parameter $\theta$ for the trajectory-conditioned policy $\pi_\theta(a_t|e_{\leq t}, o_t, g)$
Initialize the trajectory buffer $\mathcal{D} \leftarrow \emptyset$ *# Store diverse past trajectories*
Initialize set of transitions in the current episode $\mathcal{E} \leftarrow \emptyset$ *# Store current episode trajectory*
Initialize set of on-policy samples $\mathcal{F} \leftarrow \emptyset$ *# Store data for on-policy PPO update*
Initialize demonstration trajectory $g \leftarrow \emptyset$
**for** each iteration $i$ from 1 to $I$ **do**
    **for** each step $t$ **do**
        Observe $s_t = \{o_t, e_t\}$ and choose an action $a_t \sim \pi_\theta(a_t|e_{\leq t}, o_t, g)$
        Execute action $a_t$ in the environment to get $r_t, o_{t+1}, e_{t+1}$
        Store transition $\mathcal{E} \leftarrow \mathcal{E} \cup \{(o_t, e_t, a_t, r_t)\}$
        *# Positive reward if agent follows demonstration $g$*
        *# No reward after agent completes $g$ and then takes random exploration*
        Determine $r_t^{\text{DTSIL}}$ by comparing $e_{\leq t+1}$ with $g$     (Eq. 1)
        Store on-policy sample $\mathcal{F} \leftarrow \mathcal{F} \cup \{(o_t, e_t, a_t, g, r_t^{\text{DTSIL}})\}$
    **end for**
    **if** $s_{t+1}$ is terminal **then**
        $\mathcal{D} \leftarrow \text{UpdateBuffer}(\mathcal{D}, \mathcal{E})$     (Alg. 2)
        Clear current episode trajectory $\mathcal{E} \leftarrow \emptyset$
        $g \leftarrow \text{SampleDemo}(\mathcal{D}, i, I)$     (Alg. 3)
    **end if**
    $\theta \leftarrow \theta - \eta \nabla_\theta \mathcal{L}^{\text{RL}}$ *# Perform PPO update using on-policy samples*     (Eq. 2)
    Clear on-policy samples $\mathcal{F} \leftarrow \emptyset$
    $\theta \leftarrow \theta - \eta \nabla_\theta \mathcal{L}^{\text{SL}}$ *# Perform supervised learning updates using samples from $\mathcal{D}$ for $J$ times*     (Eq. 3)
**end for**

---

agent to follow a demonstration and reach the goal state even far away from the current state. Our method shares the same motivation as Duan et al. (2017) which uses an attention model over the demonstration but mainly focuses on the block stacking task. However, our architecture is simpler since it does not use an attention model over the current observation and our method is evaluated on various environments.

## 3 METHOD

The main idea of our method is to maintain a buffer of diverse trajectories collected during training and to train a trajectory-conditioned policy by leveraging reinforcement learning and supervised learning to roughly follow demonstrations sampled from the trajectory buffer. The demonstration trajectories cover diverse possible directions in the environment. Therefore, the agent is encouraged to explore beyond various visited states in the environment and gradually push its exploration frontier further. In the meantime, we can train the policy to imitate the best trajectories collected to exploit the past good experiences. We put more weights on exploration in the early stage of training, and then increases the probability of imitating the best trajectories (i.e., exploitation) as training goes on. We name our method as *Diverse Trajectory-conditioned Self-Imitation Learning* (DTSIL).

### 3.1 BACKGROUND AND NOTATION

In the standard reinforcement learning setting, at each time step $t$, an agent observes a state $s_t$, selects an action $a_t \in \mathcal{A}$, and receives a reward $r_t$ when transitioning to a next state $s_{t+1} \in \mathcal{S}$, where $\mathcal{S}$ is a set of all states and $\mathcal{A}$ is a set of all actions. The goal is to find a policy $\pi_\theta(a|s)$ parameterized by $\theta$ that maximizes the expected return $\mathbb{E}_{\pi_\theta}[\sum_{t=0}^T \gamma^t r_t]$, where $\gamma \in (0, 1]$ is a discount factor.

In our work, we assume a state $s_t$ includes the agent's observation $o_t$ (e.g., raw pixel image) and a high-level abstract state embedding $e_t$ (e.g., the agent's location in the abstract space). The embedding $e_t$ may be learnable from $o_{\leq t}$ (e.g., ADM (Choi et al., 2018) could localize the agent in Atari games), but in this work, we consider a setting where high-level embedding is provided as a part of $s_t$ [1]. A *trajectory-conditioned policy* $\pi_\theta(a_t|e_{\leq t}, o_t, g)$ (which we refer to as $\pi_\theta(\cdot|g)$ in shorthand notation) takes a sequence of state embeddings $g = \{e_1^g, e_2^g, \cdots, e_{|g|}^g\}$ as input for a demonstration, where $|g|$ is

---

[1]In many important application domains (e.g. the robotics domain), such handcrafted representation is available. Also, learning a good state representation itself is an important open question and extremely challenging especially for hard-exploration and sparse-reward environments, which is not the main focus of this work. Therefore, we assume the availability of the high-level representations as many previous works (Florensa et al., 2017; Liu et al., 2019; Ecoffet et al., 2019; Plappert et al., 2018)

the length of the trajectory $g$. A sequence of the agent's past state embeddings $e_{\leq t} = \{e_1, e_2, \cdots, e_t\}$ is provided to determine which part of the demonstration has been followed by the agent. Together with the current observation $o_t$, it helps to determine the correct action $a_t$ to accurately imitate the demonstration. Our goal here is to find a set of optimal state embedding sequence(s) $g^*$ and the policy $\pi_\theta^*(\cdot|g)$ to maximize the return: $g^*, \theta^* \triangleq \arg\max_{g,\theta} \mathbb{E}_{\pi_\theta(\cdot|g)}[\sum_{t=0}^T \gamma^t r_t]$. For robustness we may want to find multiple near-optimal embedding sequences with similar returns and a trajectory-conditioned policy for executing them. In our implementation, we train the trajectory-conditioned policy to imitate the best trajectories. Alternatively, an unconditional stochastic policy could also be trained to imitate the best trajectories, which may further improve generalization and robustness (see Appendix C.1 for more discussion and experiments).

## 3.2 Organizing Trajectory Buffer

We maintain a *trajectory buffer* $\mathcal{D} = \{(e^{(1)}, \tau^{(1)}, n^{(1)}), (e^{(2)}, \tau^{(2)}, n^{(2)}), \cdots\}$ of diverse past trajectories. For each embedding-trajectory-count tuple $(e^{(i)}, \tau^{(i)}, n^{(i)})$, $\tau^{(i)}$ is the best trajectory ending with a state with the high-level representation $e^{(i)}$, and $n^{(i)}$ is the number of times the cluster represented by this state embedding $e^{(i)}$ has been visited during training. To maintain a compact buffer, similar state embeddings within the tolerance threshold $th$ can be clustered together, and the existing entry is replaced if an improved trajectory $\tau^{(i)}$ ending with a near-identical state is found.

When given a new episode $\mathcal{E} = \{(o_0, e_0, a_0, r_0), \cdots, (o_T, e_T, a_T, r_T)\}$, all of the state embeddings $e_t(1 \leq t \leq T)$ in this episode $\mathcal{E}$ are considered as follows (similarly to Ecoffet et al. (2019)), because the buffer should maintain all of the possible paths available for future exploration to avoid missing any possibility to find an optimal solution. If the Euclidean distance between $e_t$ and any state embedding $e^{(i)}$ in the buffer is larger than $th$ (i.e $e_t$ does not belong to any existing cluster in the buffer), $(e_t, \tau_{\leq t}, 1)$ is directly pushed into the buffer, where $\tau_{\leq t} = \{(o_0, e_0, a_0, r_0), \cdots, (o_t, e_t, a_t, r_t)\}$ is the agent's partial episode ending with $e_t$. If there exists $e^{(k)}$ similar to $e_t$ (i.e., $e^{(k)}$ and $e_t$ belong to the same cluster within threshold $th$) and the partial episode $\tau_{\leq t}$ is better (i.e., higher return or shorter trajectory) than the stored trajectory $\tau^{(k)}$, $\tau^{(k)}$ is replaced by the current trajectory $\tau_{\leq t}$, and $e^{(k)}$ is replaced by $e_t$ to represent this cluster of state embeddings. The full algorithm in pseudo-code is described in Appendix A.1.

## 3.3 Sampling Demonstrations

When learning a trajectory-conditioned policy $\pi$, demonstration trajectories are sampled from the buffer $\mathcal{D}$. We record the count $n^{(i)}$ of how many times the cluster represented by this state embedding $e^{(i)}$ is visited. In the exploration mode, we set the probability of sampling each trajectory as $1/\sqrt{n^{(i)}}$. This is inspired by the count-based exploration bonus (Strehl & Littman, 2008; Bellemare et al., 2016) and the idea of rank-based prioritization (Schaul et al., 2015b; Ecoffet et al., 2019): we prioritize a trajectory that ends with a less frequently visited state because this leads the agent to reach rarely visited regions in the state space and is more promising for discovering novel states.

On the other hand, in the imitation mode, we sample the best trajectories stored in the buffer for imitation learning. These trajectories are used to train the policy to converge to a high-reward behavior (Aytar et al., 2018; Ecoffet et al., 2019). To balance between exploration and exploitation, we decrease the probability of taking the exploration mode and exploit the best experiences more as training goes on. The algorithm is described in Appendix A.2.

## 3.4 Learning Trajectory-Conditioned Policy

**Imitation Reward**   Given a demonstration trajectory $g = \{e_0^g, e_1^g, \cdots, e_{|g|}^g\}$, we provide reward signals for imitating $g$. At the beginning of an episode, the index $u$ of the last visited state embedding in the demonstration is initialized as $u = -1$. At each step $t$, if the agent's new state $s_{t+1}$ has an embedding $e_{t+1}$ and it is the similar enough to any of the next $\Delta t$ state embeddings starting from the last visited state embedding $e_u^g$ in the demonstration (i.e., $\|e_{t+1} - e_{u'}^g\| < th$ where $u < u' \leq u + \Delta t$), then it receives a positive imitation reward $r^{\text{im}}$, and the index of the last visited state embedding in the demonstration is updated as $u \leftarrow u'$. This encourages the agent to visit the state embeddings in the demonstration in a soft-order so that the agent could explore around the demonstration and the demonstration plays a role to guide the agent to the region of interest in the state embedding space. To summarize, the agent receives a reward $r_t^{\text{DTSIL}}$ defined as

$$r_t^{\text{DTSIL}} = \begin{cases} f(r_t) + r^{\text{im}} & \text{if } \exists u', u < u' \leq u + \Delta t, \text{ such that } \|e_{u'}^g - e_{t+1}\| < th \\ 0 & \text{otherwise,} \end{cases} \tag{1}$$

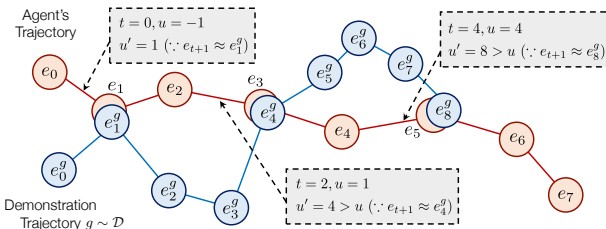

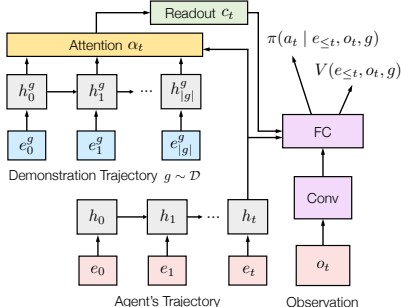

Figure 2: An example showing the updates of $u$, given $\Delta t = 4$. At each step $t$, we check the state embedding $e_{t+1}$ to find similar state embedding $e^g_{u'}$ satisfying $e_{t+1} \approx e^g_{u'}$ (i.e. $\|e_{t+1} - e^g_{u'}\| < th$) and to determine the reward according to Equation 1. After completing the demonstration, the agent performs random exploration with reward 0 ($e_5 \sim e_7$).

Figure 3: Architecture of the trajectory-conditioned policy (see Appendix B).

where $f(\cdot)$ is a monotonically increasing function (e.g., clipping (Mnih et al., 2015)). Figure 2 illustrates the updates of $u$ during an episode when the agent visits a state whose embedding is close to state embeddings in the demonstration $g$.

**Policy Architecture** For imitation learning with diverse demonstrations, we design a trajectory-conditioned policy $\pi_\theta(a_t|e_{\leq t}, o_t, g)$ that should imitate any given trajectory $g$. Inspired by neural machine translation methods (Sutskever et al., 2014; Bahdanau et al., 2014), one can view the demonstration as the source sequence and view the incomplete trajectory of the agent's state representations as the target sequence. We apply a recurrent neural network (RNN) and an attention mechanism to the sequence data to predict actions that would make the agent follow the demonstration.

As illustrated in Figure 3, RNN computes the hidden features $h^g_i$ for each state embedding $e^g_i$ ($0 \leq i \leq |g|$) in the demonstration and derives the hidden features $h_t$ for the agent's state representation $e_t$. Then the attention weight $\alpha_t$ is computed by comparing the current agent's hidden features $h_t$ with the demonstration's hidden features $h^g_i$ ($0 \leq i \leq |g|$). The attention readout $c_t$ is computed as an attention-weighted summation of the demonstration's hidden features to capture the relevant information in the demonstration trajectory and to predict the action $a_t$. The more details of policy architecture are described in Appendix B.

**Reinforcement Learning Objective** With the reward defined as $r^{\mathrm{DTSIL}}_t$ (Equation 1), the trajectory-conditioned policy $\pi_\theta$ can be trained with a policy gradient algorithm (Sutton et al., 2000):

$$\mathcal{L}^{\mathrm{RL}} = \mathbb{E}_{\pi_\theta}[-\log \pi_\theta(a_t|e_{\leq t}, o_t, g)\widehat{A}_t],$$

$$\text{where } \widehat{A}_t = \sum_{d=0}^{n-1} \gamma^d r^{\mathrm{DTSIL}}_{t+d} + \gamma^n V_\theta(e_{\leq t+n}, o_{t+n}, g) - V_\theta(e_{\leq t}, o_t, g), \tag{2}$$

where the expectation $\mathbb{E}_{\pi_\theta}$ indicates the empirical average over a finite batch of on-policy samples and $n$ denotes the number of rollout steps taken in each iteration. We use Proximal Policy Optimization (PPO) (Schulman et al., 2017) as an actor-critic policy gradient algorithm for our experiments.

**Supervised Learning Objective** To improve trajectory-conditioned imitation learning and to better leverage the past trajectories, we propose a supervised learning objective. We sample a trajectory $\tau = \{(o_0, e_0, a_0, r_0), (o_1, e_1, a_1, r_1) \cdots \} \in \mathcal{D}$, formulate the demonstration $g = \{e_0, e_1, \cdots, e_{|g|}\}$ and assume the agent's incomplete trajectory is the partial trajectory $g_{\leq t} = e_{\leq t} = \{e_0, e_1, \cdots, e_t\}$ for any $1 \leq t \leq |g|$. Then $a_t$ is the 'correct' action at step $t$ for the agent to imitate the demonstration. Our supervised learning objective is to maximize the log probability of taking such actions:

$$\mathcal{L}^{\mathrm{SL}} = -\log \pi_\theta(a_t|e_{\leq t}, o_t, g), \text{ where } g = \{e_0, e_1, \cdots, e_{|g|}\}. \tag{3}$$

## 4 EXPERIMENTS

In the experiments, we aim to answer the following questions: (1) How well does the trajectory-conditioned policy imitate the diverse demonstration trajectories? (2) Does imitation of the past diverse experience enable the agent to further explore more diverse directions and guide the exploration to find the trajectory with a near-optimal total reward? (3) Can our proposed method aid in avoiding myopic behaviors and converge to near-optimal solutions?

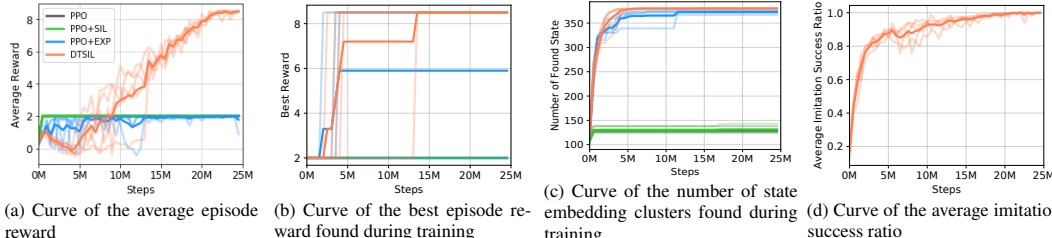

(a) Curve of the average episode reward

(b) Curve of the best episode reward found during training

(c) Curve of the number of state embedding clusters found during training

(d) Curve of the average imitation success ratio

Figure 4: Learning curves on Apple-Gold domain averaged over 5 runs, where the curves in dark colors are average over 5 curves in light colors. The x-axis and y-axis correspond to the number of steps and statistics about the performance, respectively. The average reward and average imitation success ratio are the mean values over 40 recent episodes.

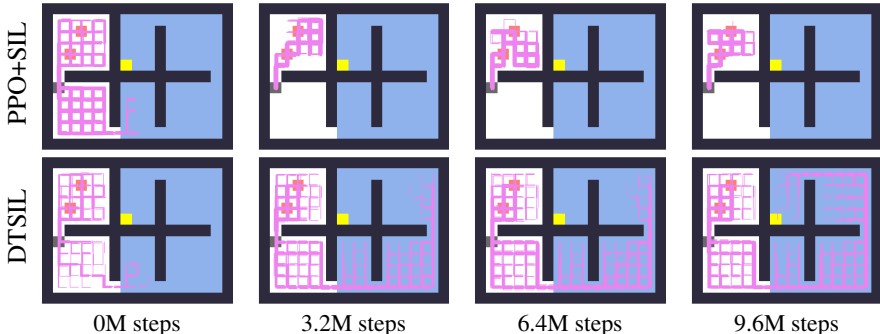

0M steps        3.2M steps        6.4M steps        9.6M steps

Figure 5: Visualization of trajectories stored in the buffer for PPO+SIL and DTSIL (ours) over time. The agent (gray), apple (red) and gold (yellow) are shown as squares for simplicity. The rocky region is in light blue.

We compare our method with the following baselines: (1) PPO: Proximal Policy Optimization (Schulman et al., 2017); (2) PPO+EXP: PPO with reward $f(r_t) + \lambda/\sqrt{N(e_t)}$, where $\lambda/\sqrt{N(e_t)}$ is the count-based exploration bonus, $N(e)$ is the number of times the cluster which the state representation $e$ belongs to was visited during training and $\lambda$ is the hyper-parameter controlling the weight of exploration term; (3) PPO+SIL: PPO with Self-Imitation Learning (Oh et al., 2018). More details about the implementation can be found in the Appendix.

## 4.1 APPLE-GOLD DOMAIN

The Apple-Gold domain (shown in Figure 1) is a simple grid-world environment with misleading rewards that can lead the agent to local optima. An observation consists of the agent's location $(x_t, y_t)$ and binary variables showing whether the agent has gotten the apple or the gold. A state is represented as the agent's location and the cumulative positive reward: $e_t = (x_t, y_t, \sum_{i=1}^{t} \max(r_i, 0))$, indicating the location of the agent and the collected objects.

As shown in Figure 4a, PPO, PPO+SIL, and PPO+EXP agents are stuck with the sub-optimal policy of collecting the two apples. In Figure 4b, PPO+EXP agent could occasionally explore further and gather the gold with total reward 8.5. However, the agent does not replicate the good trajectory due to the negative reward along the optimal path and network forgetting about the good experiences. DTSIL marches forward on the right side of the maze and achieves the highest total reward 8.5 within the time limit. Figure 4c shows the number of different state embeddings found during training.

In Figure 4d, we show the average success ratio of the imitation during training. It is defined as follows: for a given demonstration $g = \{e_0^g, e_1^g, \cdots, e_{|g|}^g\}$, let $u$ be the index of the last visited state embedding in $g$ when the agent's current episode terminates, then the success ratio of imitating $g$ is $\frac{u}{|g|}$ (i.e., the portion of trajectory imitated). Ideally, we want the success ratio to be 1.0, which indicates that the trajectory-conditioned policy could successfully follow any given demonstration from the buffer. At 5M steps, the trajectories with the optimal total reward 8.5 are found, and our trajectory-conditioned policy eventually imitates them well with a success ratio around 1.0.

Figure 5 visualizes a learning process. PPO+SIL fails on this task because the agent quickly exploits good experiences of collecting the apples and the buffer is filled with the trajectories in the

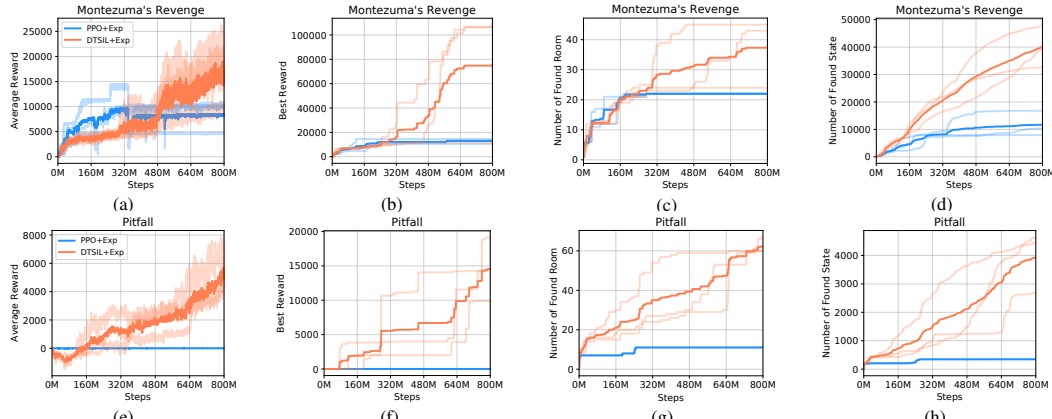

Figure 6: Learning curves of the average episode reward, the best episode reward, and the number of different rooms found on Montezuma's Revenge and Pitfall, averaged over 3 runs. On Montezuma's Revenge, DT-SIL+EXP discovers around 40 rooms on average while PPO+EXP never finds a path to pass through all 24 rooms at the first level and then proceed to the next level.

nearby region. On the contrary, DTSIL maintains a buffer of diverse trajectories which are used as demonstrations to guide the agent to explore different regions and discover an optimal behavior.

### 4.2 ATARI MONTEZUMA'S REVENGE AND PITFALL

We evaluate our method on the hard-exploration game Montezuma's Revenge and Pitfall in the Arcade Learning Environment (ALE) (Bellemare et al., 2013; Machado et al., 2017). The environment setting is the same as Mnih et al. (2015). There is a random initial delay resulting in stochasticity in the environment. The observation is a frame of raw pixel images, and the state representation $e_t = (\text{room}_t, x_t, y_t, \sum_{i=1}^{t} \max(r_i, 0))$ consists of the agent's ground truth location (obtained from RAM) and the accumulated positive environment reward, which implicitly indicates the objects the agent has collected[2]. It is worth noting that even with the ground-truth location of the agent, on these two infamously difficult games, it is highly non-trivial to explore efficiently and avoid local optima without relying on expert demonstrations or being able to reset to arbitrary states. In addition to the agent's location information, many complicated elements such as moving entities, traps, and the agent's inventory are included in the state. Therefore, these Atari games with agent's location information are still much more challenging than the grid world environments. Empirically, as summarized in Table 1, the previous SOTA baselines using the agent's ground truth location information even fails to achieve high scores.

Using the state representation $e_t$, we introduce a variant 'DTSIL+EXP' that adds a count-based exploration bonus $r_t^+ = 1/\sqrt{N(e_t)}$ to Eq.1 for faster exploration[3]. As shown in Figure 6a and 6e, in the early stage, the average episode reward of DTSIL+EXP is worse than PPO+EXP because our policy is trained to imitate diverse demonstrations rather than directly maximize the environment reward. Contrary to PPO+EXP, DTSIL+EXP agent is not eager to myopically follow the high-reward path since the path with a relatively low score in the short term might lead to higher rewards in the long term. On Montezuma's Revenge, for example, with two keys in hand, PPO+EXP agent often opens a nearby door and loses the chance of opening the last two doors of the first level[4]. As training continues, DTSIL+EXP successfully discovers trajectories to pass the first level with a total reward of more than 20,000, as shown in Figure 6b. While gradually increasing the probability of imitating the best trajectories in the buffer by sampling them as demonstrations, the average episode reward

---

[2]We can also use the number of keys as an element in the state embedding as in (Ecoffet et al., 2019) to reduce the size of the state embedding space and improve the performance, as shown in Appendix H.

[3]Note that the existing exploration methods listed in Table 1 already take advantage of count-based exploration bonus (e.g., A2C+CoEX+RAM, SmartHash, DeepCS, and Abstract-HRL). Therefore, combination of DTSIL and the count-based exploration bonus does not introduce unfair advantages over other baselines.

[4]Demo videos of the learned policies for both PPO+EXP and DTSIL+EXP are available at https://sites.google.com/view/diverse-sil. In comparison to DTSIL+EXP, we could see the PPO+EXP agent does not explore enough to make best use of the tools (e.g. sword, key) collected in the game. A map of this level is shown in Figure 16 in Appendix.

| Method | DTSIL | A2C+CoEX+RAM | SmartHash | DeepCS | Abstract-HRL | A2C+SIL | PPO+CoEX | RND |
|---|---|---|---|---|---|---|---|---|
| MontezumaRevenge | **20,187** | 6,600 | 5,661 | 3,500 | 11,000 | 2,500 | 11,618 | 10,070 |
| Pitfall | **6,546** | - | - | -186 | 10,000 | - | - | -3 |

Table 1: Comparison with the state-of-the-art results on Montezuma's Revenge and Pitfall. Abstract-HRL Liu et al. (2019) assumes more high-level state information, including the agent's location, inventory and invetory history, etc. DTSIL, A2C+CoEX+RAM (Choi et al., 2018), SmartHash (Tang et al., 2017), and DeepCS (Stanton & Clune, 2018) only make use of agent's location information from RAM, while A2C+SIL (Oh et al., 2018), PPO+CoEX (Choi et al., 2018), and RND (Burda et al., 2018) do not use RAM information. The score is averaged over multiple runs, gathered from each paper.

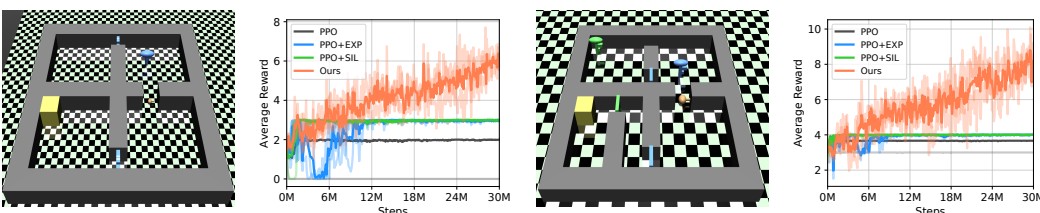

Figure 7: The reward for getting the key, opening the door, and collecting the treasure (yellow block) is 1, 2, and 6 respectively. The learning curve of the episode reward is averaged over 3 independent runs.

could increase to surpass 20,000 in Figure 6a. On Pitfall, the positive reward is much sparser and most of the actions yield small negative rewards that would discourage getting a high total reward in the long term. However, our method still stores the trajectories with negative rewards, encourages the agent to visit these novel regions and then discovers good paths with positive rewards as illustrated in Figure 6f. Therefore we are able to eventually reach average episode reward over 0 in Figure 6e, without expert demonstrations.

Table 1 compares our proposed method with previous works without using any expert demonstration or resetting to an arbitrary state, where our approach significantly outperforms the other approaches which make use of the same information from RAM about the agent's location. In the Appendix C.3, we present more experimental results on other interesting environments with discrete action space such as Deep Sea (Osband et al., 2019).

## 4.3 MuJoCo

We evaluate DTSIL on continuous control tasks. We adapt the maze environment introduced in (Duan et al., 2016) to construct a set of challenging tasks, which require the point mass agent to collect the key, open the door with the same color and finally reach the treasure to get a high score. One key cannot be re-used once it was used before to open a door with the same color, which makes the agent to be easily trapped. A visualization of these environments is shown in Figure 7. The agent's initial location is randomly sampled from a Gaussian distribution as in standard MuJoco tasks (Brockman et al., 2016). The observation is the agent's location and range sensor reading about nearby objects. The state representation is $e_t = (x_t, y_t, \sum_{i=1}^{t} r_i)$.

As shown in the first maze of Figure 7, the agent can easily get the blue key near its initial location and open the blue door in the upper part. However, the optimal path is to bring the key to open the blue door in the bottom and obtain the treasure, reaching an episode reward of 9. In the second maze, the agent should bring the blue key and pick up the green key while avoiding opening the blue door in the upper part. Then, the green and blue key can open the two doors at the bottom of the maze, which results in the total reward of 12. The learning curves in Figure 7 show that PPO, PPO+EXP, and PPO+SIL may get stuck at a sub-optimal behavior, whereas our policy eventually converges to the behavior achieving the high episode reward.

## 5 Conclusion

This paper proposes to learn diverse policies by imitating diverse trajectory-level demonstrations through count-based exploration over these trajectories. Imitation of diverse past trajectories can guide the agent to rarely visited states and encourages further exploration of novel states. We show that in a variety of environments with local optima, our method significantly improves self-imitation learning (SIL). It avoids prematurely converging to a myopic solution and learns a near-optimal behavior to achieve a high total reward.

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

# Appendix

## A DETAILED DESCRIPTION OF ALGORITHMS

### A.1 ALGORITHM OF TRAJECTORY BUFFER UPDATE

In Algorithm 2, we summarize how to process the collected episode and store the diverse trajectories in the trajectory buffer.

---
**Algorithm 2** Update Trajectory Buffer

---

Input: the trajectory buffer $\mathcal{D} = \{(e^{(1)}, \tau^{(1)}, n^{(1)}), (e^{(2)}, \tau^{(2)}, n^{(2)}), \cdots\}$
Input: the current episode $\mathcal{E} = \{(o_0, e_0, a_0, r_0), (o_1, e_1, a_1, r_1), \cdots, (o_T, e_T, a_T, r_T)\}$
Input: the threshold $th$ for high level state embedding

***# Consider all the states in $\mathcal{E}$***
**for** each step $t$ **do**
  ***# Consider state $s_t$ and partial episode*** $\tau_{\leq t} = \{(o_0, e_0, a_0, r_0), \cdots, (o_t, e_t, a_t, r_t)\}$
  **if** there exists $(e^{(k)}, \tau^{(k)}, n^{(k)}) \in \mathcal{D}$ where $\|e^{(k)} - e_t\| < th$ **then**
    ***# Compare partial episode $\tau_{\leq t}$ with stored trajectory $\tau^{(k)}$***
    **if** $\tau_{\leq t}$ has higher total reward or reaches the same total reward with less steps **then**
      $\tau^{(k)} \leftarrow \tau_{\leq t} = \{(o_0, e_0, a_0, r_0), (o_1, e_1, a_1, r_1), \cdots, (o_t, e_t, a_t, r_t)\}$
      $e^{(k)} \leftarrow e_t$
    **end if**
    $n^{(k)} \leftarrow n^{(k)} + 1$
  **else**
    $\mathcal{D} \leftarrow \mathcal{D} \cup (e_t, \tau_{\leq t}, 1)$ where $\tau_{\leq t} = \{(o_0, e_0, a_0, r_0), (o_1, e_1, a_1, r_1), \cdots, (o_t, e_t, a_t, r_t)\}$
  **end if**
**end for**
**return** $\mathcal{D}$

---

### A.2 ALGORITHM OF SAMPLING DEMONSTRATIONS

In Algorithm 3, we summarize how to sample the demonstrations from the trajectory buffer for exploration or imitation. Considering the current iteration $i$ and the total number of iterations $I$, the probability of sampling demonstration for imitation to learn good behavior is $\frac{i}{I}$ and the probability of sampling demonstration from exploration is $1 - \frac{i}{I}$.

---
**Algorithm 3** Sample Demonstration Trajectories

---

Input: the trajectory buffer $\mathcal{D} = \{e^{(1)}, \tau^{(1)}, n^{(1)}), (e^{(2)}, \tau^{(2)}, n^{(2)}), \cdots\}$
Input: current iteration $i$, total number of iterations $I$.

***# With probability $\frac{i}{I}$, run the imitation mode; with probability $1 - \frac{i}{I}$, run the exploration mode***
**if** random number $\sim U[0, 1]$ is smaller than $\frac{i}{I}$ **then**
  ***# sample the top-K trajectories reaching near-optimal score in the buffer***
  $g \leftarrow \{e_0, e_1, \cdots, e_{|g|}\}$ for all $(o_t, e_t, a_t, r_t) \in \tau^{best}$
**else**
  Calculate probability distribution $p \leftarrow [\frac{1}{\sqrt{n^{(1)}}}, \frac{1}{\sqrt{n^{(2)}}}, \cdots]$
  $p \leftarrow \frac{p}{\sum_j p_j}$
  Sample $(e, \tau, n) \sim \text{Categorical}(\mathcal{D}, p)$
  $g \leftarrow \{e_0, e_1, \cdots, e_{|g|}\}$ for all $(o_t, e_t, a_t, r_t) \in \tau$
**end if**
**return** $g$

---

## B  DETAILS OF NETWORK ARCHITECTURE AND TRAINING PROCESS

In the trajectory-conditioned policy (Figure 3), we first embed the input state $e_t$ (or $e_i^g$) with a fully-connected layer with 64 units. Next, a RNN with gated recurrent units (GRU) computes the feature $h_t$ (or $h_i^g$) with 128 units. The attention weight $\alpha_t$ is calculated based on the Bahdanau attention mechanism (Bahdanau et al., 2014). The concatenation of the attention readout $c_t$, the hidden feature of agent's current state $h_t$, and convolutional features from the observation are used to predict $\pi(a_t|e_{\leq t}, o_t, g)$ with a linear layer.

For experiments on the Apple-Key domain, Toy Montezuma's Revenge, and Mujoco, the features from $o_t$ are not required for the policy. However, on the Atari games such as Montezuma's Revenge, it is necessary to take the raw observation $o_t$ as input into policy because the location information in $e_{\leq t}$ solely could not let the agent to take temporal context into account (e.g. avoiding moving skulls and passing laser gates). With the raw observation $o_t$ with shape $84 \times 84 \times 4$ as input, three convolutional layers are used to encode $o_t$ and then the convolutional feature is flattened.

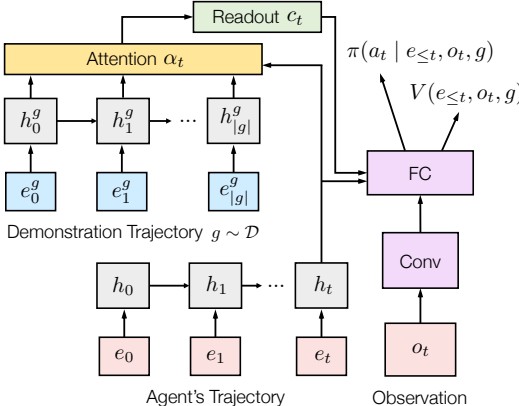

Figure 8: Architecture of the trajectory-conditioned policy (Repeating Figure 3).

During training, our algorithm begins with an empty buffer $\mathcal{D}$. We initialize the demonstration as a list of zero vectors. With such an input demonstration, the agent performs random exploration to collect trajectories to fill the buffer $\mathcal{D}$. In practice, the sampled demonstration trajectory $g = \{e_0^g, e_1^g, \cdots, e_{|g|}^g\}$ could be lengthy. We present a part of the demonstration as the input into the policy, similarly to translating a paragraph sentence by sentence. Specifically, we first input $\{e_0^g, e_1^g, \cdots, e_m^g\}$ ($m \leq |g|$) into the policy. When the index of the agent's last visited state embedding in the demonstration $u$ belongs to $\{m - \Delta t, \cdots, m\}$, we think that the agent has accomplished this part of the demonstration, and switch to the next part $\{e_u^g, e_{u+1}^g, \cdots, e_{u+m}^g\}$. We repeat this process until the last part of the demonstration. If the last part $\{e_u^g, e_{u+1}^g, \cdots, e_{|g|}^g\}$ is less than $m + 1$ steps long, we pad the sequence with zero vectors.

A reward function $f(r_t) = r_t$ is used on the Apple-Gold, Deep Sea and MuJoco domain, and $f(r_t) = 2 \cdot \text{clip}(r_t, 0, 1)$ on other environments. $r^{\text{im}} = 0.1$ is the reward to encourage imitation. More details about hyperparameters and the environments can be found in the Appendix D.

# C  ADDITIONAL EXPERIMENTS

## C.1  GENERALIZATION AND ROBUSTNESS IN STOCHASTIC ENVIRONMENTS

We evaluate our method on environments with different levels of stochasticity. For Apple-Gold domain, in the environments with random initial location of the agent (Figure 9), or with sticky action (Figure 10), our DTSIL still outperforms the baselines and achieves near-optimal total episode reward.

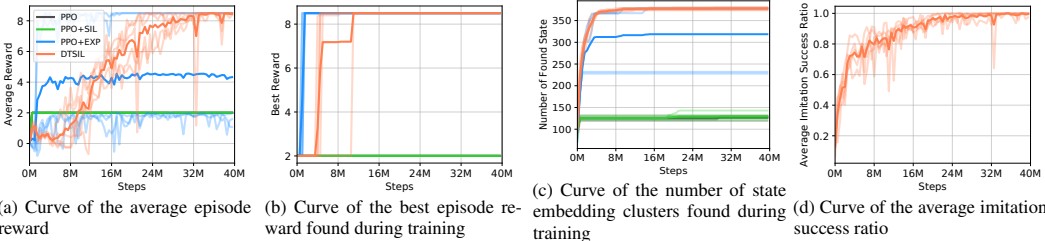

(a) Curve of the average episode reward

(b) Curve of the best episode reward found during training

(c) Curve of the number of state embedding clusters found during training

(d) Curve of the average imitation success ratio

Figure 9: Learning curves on Apple-Gold domain with random initial location of the agent in the lower left corner, where the curves in dark colors are average over 5 curves in light colors. The x-axis and y-axis correspond to the number of steps and statistics about the performance, respectively. The average reward and average imitation success ratio are the mean values over 40 recent episodes.

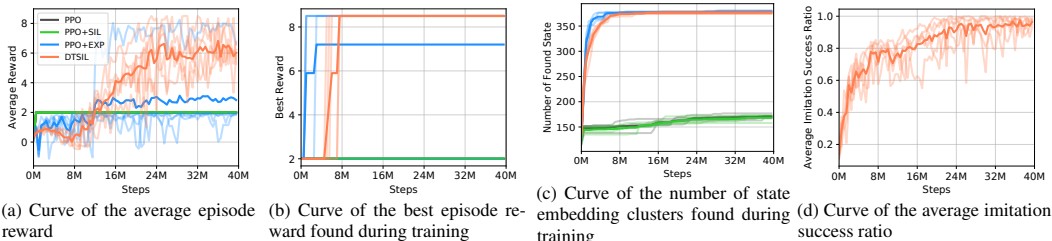

(a) Curve of the average episode reward

(b) Curve of the best episode reward found during training

(c) Curve of the number of state embedding clusters found during training

(d) Curve of the average imitation success ratio

Figure 10: Learning curves on Apple-Gold domain with sticky action of the agent, where the curves in dark colors are average over 5 curves in light colors. The x-axis and y-axis correspond to the number of steps and statistics about the performance, respectively. The average reward and average imitation success ratio are the mean values over 40 recent episodes.

In a more challenging scenario when the location of the objects can be random, previous works using expert demonstrations (e.g. Aytar et al. (2018)) would also struggle. Our method can be easily extended to handle this difficulty by ditilling the behavior of good trajectories we collected to train an unconditional policy robust to the stochasticity. For example, on the Apple-Gold domain with pixel observation (Figure 11), the location of the gold could be random in the upper middle part of the maze. We first explore for a sufficiently large number of timesteps (e.g. 10M timesteps) with the trajectory-conditioned policy to collect good trajectories and then train an unconditional policy by distilling the behavior, as shown in Figure 11, to always collect the gold. We could see DTSIL with unconditional policy training is able to generalize in the stochastic environment.

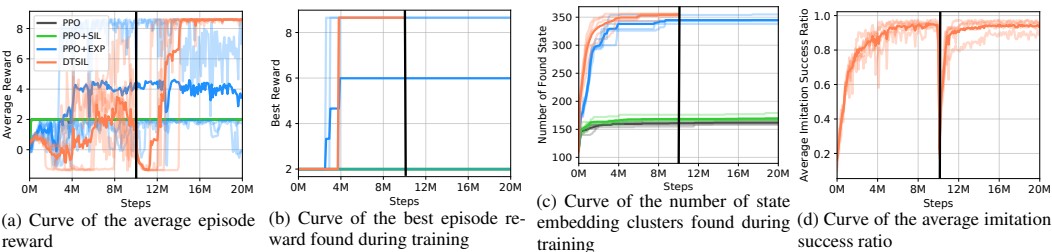

(a) Curve of the average episode reward

(b) Curve of the best episode reward found during training

(c) Curve of the number of state embedding clusters found during training

(d) Curve of the average imitation success ratio

Figure 11: Learning curves on Apple-Gold domain with stochastic location of the gold in the upper middle part of the maze, where the curves in dark colors are average over 5 curves in light colors. The x-axis and y-axis correspond to the number of steps and statistics about the performance, respectively. The average reward and average imitation success ratio are the mean values over 40 recent episodes.

## C.2 EXPERIMENTS ON TOY MONTEZUMAREVENGE

We evaluate our method on a more challenging domain, Toy Montezuma's Revenge (Roderick et al., 2018), which requires a more sophisticated strategy to explore the environment. As shown in Figure 12, there are 24 rooms similar to the layout of the first level of Atari Montezuma's Revenge, with a discrete grid for each room. The agent should navigate the labyrinth to locate the keys, unlock the doors and reach the goal (the treasure room). The observation is represented by the agent's location and cumulative episode reward. The state representation $e_t = (\text{room}_t, x_t, y_t, \sum_{i=1}^{t} r_i)$ is the same as the observation.

The learning curve of the averaged episode reward in Figure 13 shows that PPO, PPO+SIL, and PPO+EXP could not learn a policy to reach the goal. The PPO+EXP agent occasionally finds a trajectory with the total reward of 11,200 reaching the treasure room, but fails to exploit this experience. On the other hand, our method learns a good behavior of not only reaching the goal room, but also collecting all of the keys to achieve an optimal total reward of 11,600.

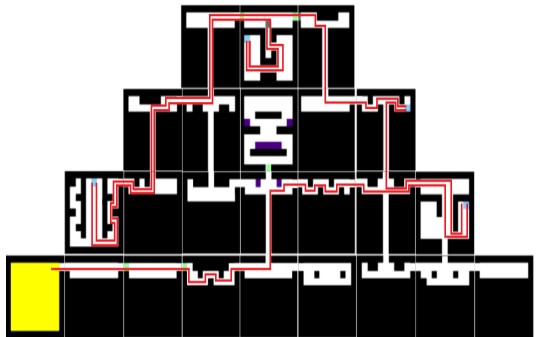

Figure 12: Map of Toy Montezuma's Revenge, where we show the agent (gray), key(blue), door(green), and treasure (yellow) as squares. The rewards are 100, 300, and 10000, respectively. An optimal path with the highest total reward of 11,600 is shown as a red line.

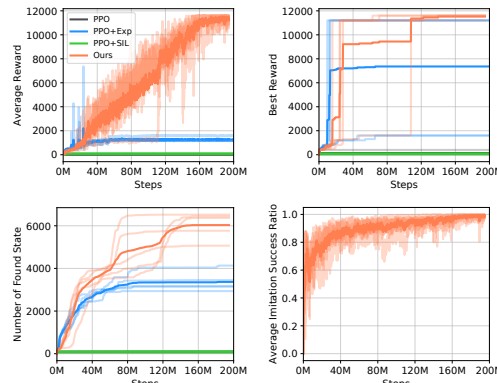

Figure 13: Learning curves on Toy Montezuma's Revenge averaged over 5 runs.

## C.3 EXPERIMENTS ON DEEP SEA

As introduced in Osband et al. (2019), the deep sea problem is implemented as an $N \times N$ grid with a one-hot encoding for state. The agent begins each episode in the top left corner of the grid and descends one row per timestep. Each episode terminates after N steps, when the agent reaches the bottom row. In each state there is a random but fixed mapping between actions $A = \{0, 1\}$ and the transitions 'left' and 'right'. At each timestep there is a small cost $r = -0.01/N$ of moving right, and $r = 0$ for moving left. However, should the agent transition right at every timestep of the episode it will be rewarded with an additional reward of +1. This presents a particularly challenging exploration problem for two reasons. First, following the 'gradient' of small intermediate rewards leads the agent away from the optimal policy. Second, a policy that explores with actions uniformly at random has probability $2^{-N}$ of reaching the rewarding state in any episode.

We compare DTSIL and baselines on deep sea environments with $10 \times 10$ grid and and $30 \times 30$ grid. The state embedding we use here is exactly the observation. The result is shown in Figure 14. On the first environment, it is easy for all of the methods to converge to the optimal behavior. The second one is much more challenging to find the optimal trajectory maximizing total reward. Therefore, PPO and PPO+SIL fails at such environment due to the hard exploration. PPO+EXP could not always explore to find the good behavior and exploit it efficiently within 12M timesteps. DTSIL successfully discovers the right way and imitate to converge to the optimal behavior.

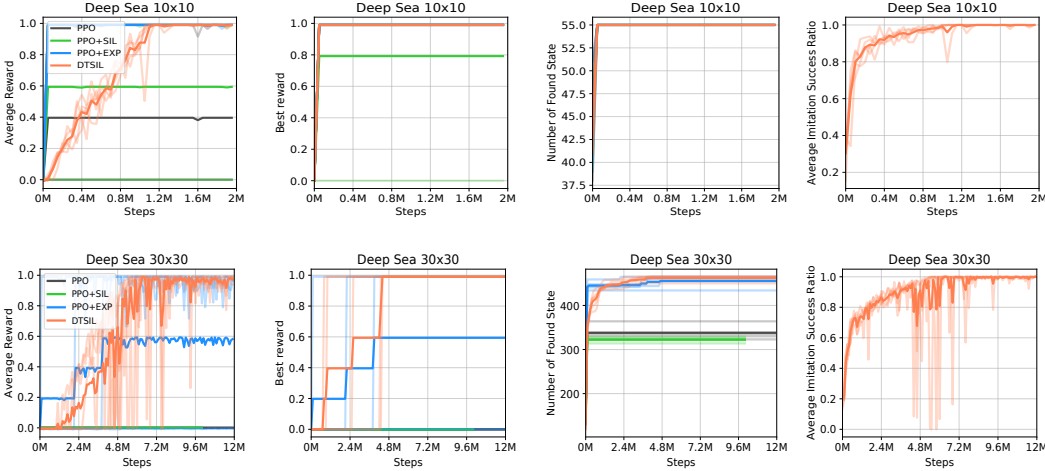

Figure 14: Experiment on Deep Sea. The learning curves shows the average episode reward, best episode reward, the number of found state representations, and the average success ratio of imitating the demonstrations in order. The curves are averaged over 5 independent runs.

# D  HYPERPARAMETERS

The hyper-parameters for our proposed method used in each experiment are listed in Table 2. On Mujoco environment, RL loss alone worked well so we did not include SL loss for behavior cloning. On the other environments when action prediction in behavior cloning is poor, we set a large $J$ for quickly learning to imitate demonstrations. When action prediction is accurate enough, we de-emphasize behavior cloning to enhance exploration around the demonstration. $\Delta t$ influences how flexibly the demonstration should be followed. In our experiment, we have $\Delta t < m$ due to the limit of the length $m$ of the input demonstration part. When the demonstration is longer and harder to follow, we would want larger $\Delta t$ to generously provide imitation reward (More detailed ablation study of the hyper-parameter $\Delta_t$ is in Appendix J). On Atari games, there are much more different trajectories stored in the buffer, so we sample top-100 trajectories as demonstration for imitation of best experiences. On the other environments, the total number of trajectories is much smaller, so we only take top-10 or top-1.

| Environment | Apple-Gold | Toy MontezumaRevenge | Atari | Deep Sea | Mujoco |
|---|---|---|---|---|---|
| Learning Rate $\eta$ | 2.5e-4 | 2.5e-4 | 2.5e-4 | 2.5e-4 | 1e-4 |
| $\Delta t$ | 2 | 4 | 8 | 2 | 8 |
| Length of demonstration input part $m$ | 10 | 10 | 10 | 10 | 10 |
| Number of supervised learning updates $J$ | 10 decreases to 1 when action prediction accuracy $\geq 0.75$ | 10 decreases to 1 when action prediction accuracy $\geq 0.75$ | 10 decreases to 1 when action prediction accuracy $\geq 0.75$ | 10 decreases to 1 when action prediction accuracy $\geq 0.75$ | 0 |
| Threshold $th$ for state embedding | 1 | 1 | 1 | 1 | 1 |
| Top-$K$ trajectories imitation | 1 | 10 | 100 | 1 | 10 |
| Weight of exploration $\lambda$ in PPO+EXP | 10 (best one among 5, 10, 20, 50) | 1 (best one among 0.5, 1, 2,4) | 1 (best one among 0.5, 1, 2,4) | 0.2 (best one among 0.1, 0.2, 0.5, 1) | 1 (best one among 0.5, 1, 2, 4) |
| Discounting factor $\gamma$ | 0.99 | 0.99 | 0.99 | 0.99 | 0.99 |

Table 2: Hyper-parameters on various environments for our experiments.

# E  ENVIRONMENT SETTING

For each experiment we conducted, we list the detailed environment setting in Table 3. There is stochasticity in the environments of Apple-Gold domain, Toy MontezumaRevenge, Atari, and Mujoco. On Atari games, we use setting of the random initial delay introduced in (Mnih et al., 2015). On Mujoco domain, the agent's initial location is randomly sampled from a Gaussian distribution, as in standard MuJoco tasks in OpenAI Gym (Brockman et al., 2016).

| Environment | Apple-Gold | Toy MontezumaRevenge | Atari | Deep Sea | Mujoco |
|---|---|---|---|---|---|
| Observation | agent's location (x, y) in 17x13 grid and binary variables indicating whether apple or gold is collected | agent's location (room, x, y) in 24x11x11 grid and accumulated reward | stacked most recent 4 gray observations with shape 84x84x4 | one-hot encoding of state in 10x10 or 30x30 grid | agent's location (x,y) in 22x22 space and range sensor reading about nearby objects |
| State Representation | $(x_t, y_t, \sum_{i=1}^{t} \max(r_i, 0))$ | $(room_t, x_t, y_t, \sum_{i=1}^{t} r_i)$ | $(room_t, x_t, y_t, \sum_{i=1}^{t} \max(r_i, 0))$ where $(x_t, y_t)$ in $9 \times 9$ grid | same as observation | $(x_t, y_t, \sum_{i=1}^{t} r_i)$ |
| Action | 5 discrete actions: up, down, left, right, noop | 5 discrete actions: up, down, left, right, noop | 18 discrete actions: noop, fire, left, $\cdots$ | 2 discrete actions: left, right | $(dx, dy)$ in continuous action space |
| Reward | rock -0.05 gold +10 apple +1 | key +100 door +300 treasure +10000 | mostly zero, sparse positive rewards when collecting objects | 0 if going left $-\frac{0.01}{10}$ or $-\frac{0.01}{30}$ if going right 1 at the last step if always going right | key +1 door +2 treasure +6 |
| Time limit | 45 steps | 1000 steps | 4500 steps | 10 or 30 steps | 1000 steps |
| Stochasticity | deterministic or take 3 random steps before the episode starts or sticky action | take 5 random steps before the episode starts | {MontezumaRevenge, Pitfall}-NoFrameskip-v4; take a random number between 0 and 30 of noop actions before the episode starts | deterministic | take random normal noise from the agent's initial position |

Table 3: The setting on various environments for our experiments.

# F   COMPARISON WITH LEARNING DIVERSE POLICIES BY SVPG

While the code for the Stein variational policy gradient (SVPG) in Gangwani et al. (2018) has not yet been released, we replicate the method in Gangwani et al. (2018) to learn diverse policies. Their experiments focus on continuous control tasks with relatively simple observation spaces with limited local optimal branches in the state space. We learn 8 diverse policies in parallel following their method on our Apple-Gold domain with discrete action space. Figure 15 shows a visualization of the learning progress: the 8 policies learn to cover different regions of the environment. The method explores better than PPO+SIL, but the exploration of each individual agent is not strong enough to find the optimal path to achieve the highest episode reward.

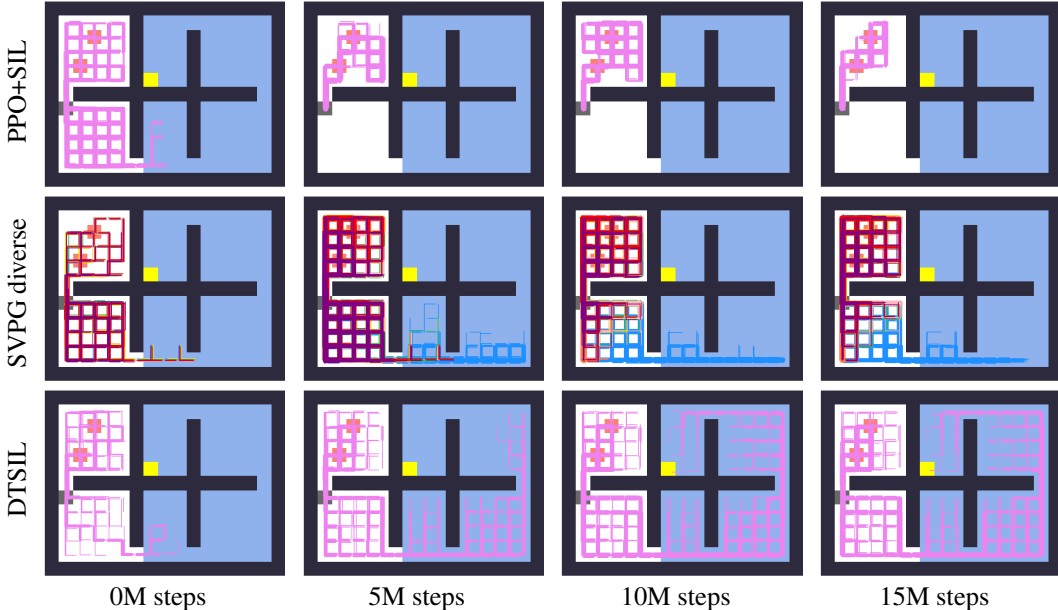

Figure 15: Visualization of the trajectories stored in the buffer for PPO+SIL, SVPG diverse Gangwani et al. (2018) and our method as training continues. In the second row, we show the trajectories for a total of 8 policies learned simultaneously with the SVPG method proposed in Gangwani et al. (2018), where each color corresponds to the trajectories collected by each policy.

## G MAP OF ATARI MONTEZUMA'S REVENGE AT THE FIRST LEVEL

On Montezuma's Revenge, there are multiple levels and each level consists of 24 rooms. A map of Atari Montezuma's Revenge at the first level is shown in Figure 16. It is challenging to bring two keys to open the two doors in room 17 behind the treasure in room 15, where the agent can pass to the next level.

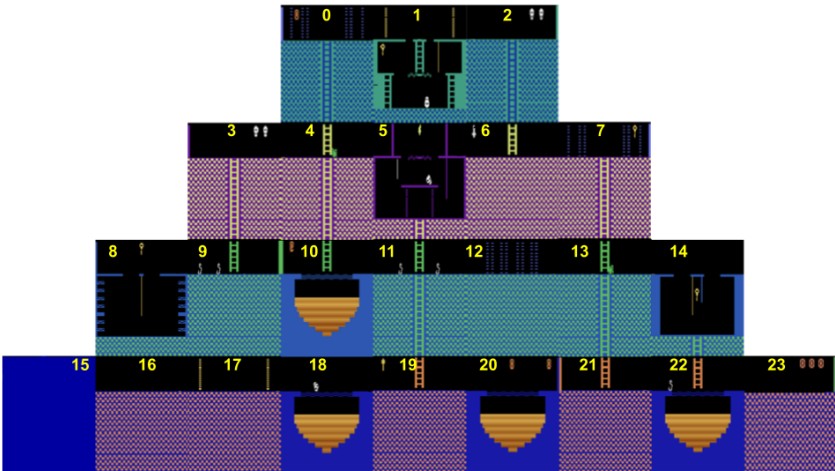

Figure 16: Map of Atari Montezuma's Revenge at the first level with 24 rooms. On Montezuma's Revenge, there are multiple levels and each level consists of 24 rooms. At the first level, it is challenging to bring two keys to open the two doors in room 17 behind the treasure in room 15, where the agent can pass to the next level.

# H  STUDY OF EXPLORATION EFFICIENCY

To evaluate the efficiency of exploration, we compare our method with the "exploration phase" in the Go-Explore algorithm (Ecoffet et al., 2019). The idea behind Go-Explore is to reset the agent to any interesting state sampled from the buffer of state embeddings, and then explore further using random actions. To study the exploration efficiency of our method, we modify the Go-Explore code such that we could not reset to any arbitrary states in the environment. Similarly to (Ecoffet et al., 2019), we use the state representation $(\text{level}_t, \text{room}_t, x_t, y_t, k_t)$ where $k_t$ is the number of keys the agent holds and $(x_t, y_t)$ is in a $9 \times 9$ grid division of the frame, and the sampling weight $\frac{1}{\sqrt{n^{(i)}}}$ to sample goal states from the buffer (It is worth noting that the state representation and goal-state sampling function recommended in Go-Explore paper is more complicated than this setting).

In the Go-Explore method without using the direct 'reset' function and with a perfect goal-conditioned policy to visit any state sampled from the buffer, the agent could precisely advance to the goal state by following the stored trajectory. The total steps taken in the environment are counted by summing the number of steps taken to follow the stored trajectories and the number of steps taken to explore.

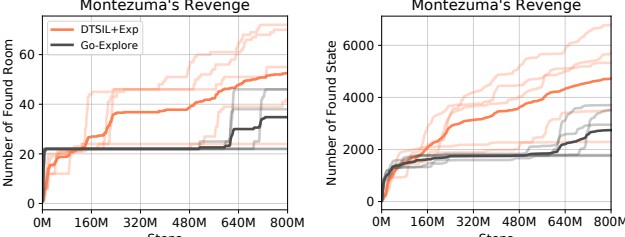

Figure 17: Learning curves of the number of rooms and the number of different state representations found on Atari Montezuma's Revenge, averaged over 5 runs. The curves in dark colors are the average of the 5 curves in light colors. During training, the state representation used is $(\text{level}_t, \text{room}_t, x_t, y_t, k_t)$.

In Figure 18, we show the average number of rooms found and the number of different state representations found during training. Even if we assume that there is a perfect goal-conditioned policy in Go-Explore to guide the agent to follow the stored trajectory exactly and visit the goal state, the learning curves demonstrate that our method is more efficient for exploring diverse state representations and consequently visits several rooms. This is because our method uses the count-based exploration bonus to encourages the exploration around and beyond the stored trajectories and the imitation reward allows the agent to follow the demonstrations in a soft-order.

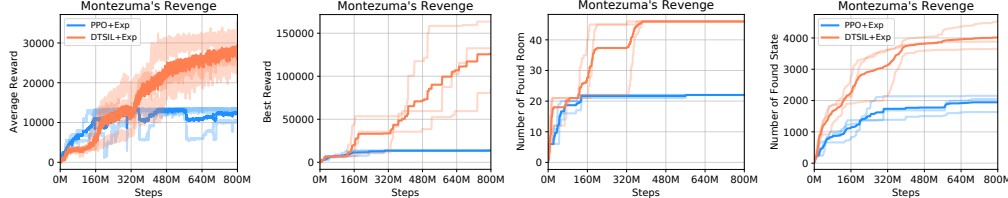

Figure 18: Learning curves of the average reward, the best episode reward, the number of rooms, the number of different state representations found on Atari Montezuma's Revenge, averaged over 5 runs. The curves in dark colors are the average of the 5 curves in light colors. During training, the state representation used is $(\text{level}_t, \text{room}_t, x_t, y_t, k_t)$.

In addition, we notice that, comparing with the state embedding $(\text{room}_t, x_t, y_t, \sum_{i=1}^{t} r_i)$, the state embedding $(\text{level}_t, \text{room}_t, x_t, y_t, k_t)$ makes the size of embedding space smaller so that the exploration could be more efficient. Such state representation conflates similar states while not conflating states that are meaningfully different. Therefore, our method could reach a higher average score around 29,817. Here, the baseline PPO+Exp is essentially the CoEX method introduced in Choi et al. (2018) with state embedding extracted from RAM, therefore DTSIL performs better than PPO+CoEX+RAM no matter the state embedding is $(\text{room}_t, x_t, y_t, \sum_{i=1}^{t} r_i)$ or $(\text{level}_t, \text{room}_t, x_t, y_t, k_t)$.

# I  STUDY OF ADVANTAGE IN EXPLOITATION

We compared our method DTSIL with PPO+EXP in the main text. PPO+EXP encourages exploration to novel states by providing auxiliary rewards to the agent, while our method rewards the agent when it successfully follow the demonstrations which leads to novel states. In order to understand more about the difference between these two mechanism, we propose a variant of our method denoted as "DTSIL-combine". In this variant, we do not separate the exploration mode and imitation mode. However, we always sample the top-K best trajectories with highest total reward $\sum_{t=0}^{|\tau|}(f(r_t) + \frac{\lambda}{\sqrt{N(e_t)}})$. The PPO+EXP baseline directly optimize the objective $\sum_{t=0}^{|\tau|}(f(r_t) + \frac{\lambda}{\sqrt{N(e_t)}})$ via the reinforcement learning algorithm while this variant indirectly optimizes such objective by imitating the best trajectories with highest value of $\sum_{t=0}^{|\tau|}(f(r_t) + \frac{\lambda}{\sqrt{N(e_t)}})$. We investigate the different performance of these two methods on the Apple-Gold domain.

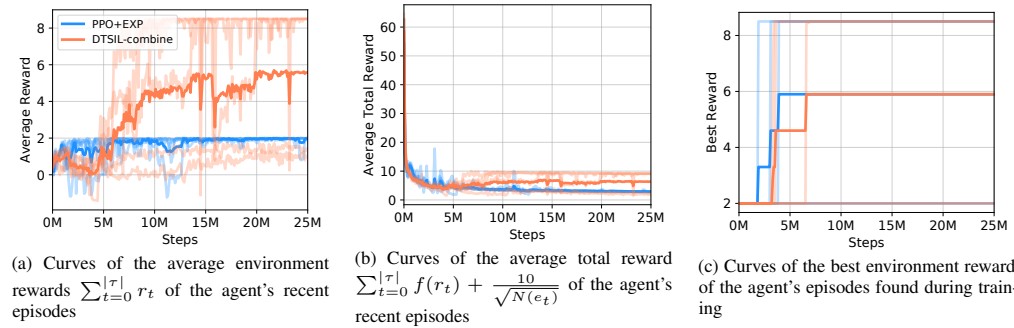

(a) Curves of the average environment rewards $\sum_{t=0}^{|\tau|} r_t$ of the agent's recent episodes

(b) Curves of the average total reward $\sum_{t=0}^{|\tau|} f(r_t) + \frac{10}{\sqrt{N(e_t)}}$ of the agent's recent episodes

(c) Curves of the best environment reward of the agent's episodes found during training

Figure 19: Learning curves averaged over 5 runs. The curves in dark colors are the average of the 5 curves in light colors.

With $\lambda = 10$ which is the best hyper-parameter we found after searching $\lambda = 5, 10, 20, 50$ for PPO+EXP on Apple-Gold domain, in Figure 19c we could notice both PPO+EXP agent and DTSIL-combine agent have 3 out of 5 runs finding the optimal trajectory with episode rewards 8.5 and the other 2 runs get stuck at the sub-optimal behavior. However, in Figure 19b, DTSIL-combine is better at optimizing the objective $\sum_{t=0}^{|\tau|} f(r_t) + \frac{10}{\sqrt{N(e_t)}}$ averaging over the agent's recent episodes and therefore it achieves higher environment environment reward as training goes on. As shown in Figure 19a, DTSIL-combine agent reproduce the good trajectories to collect the gold in 3 out of 5 runs while PPO+EXP agent is trapped at the behavior collecting the apples. The main reason might be that DTSIL-combine agent never forgets the good experience of collecting gold, and always select such good trajectories as demonstrations to guide the agent, while PPO+EXP might forget the good trajectories occasionally found or fails to exploit them before the exploration bonus vanishes. The importance of exploitation of the good experience to help the agent reproduce high-reward trajectories is also discussed in Oh et al. (2018).

## J   ABLATION STUDY OF HYPER-PARAMETER $\Delta t$

We study the effect of the hyper-parameter $\Delta t$ in the various environments. At each step, we provide $m = 10$ (considering the computational burden, we selected the value of $m$ as 10 for all experiments) steps of state embeddings from the demonstration trajectory as input into the trajectory-conditioned policy. Then we evaluate whether the agent has visited any of the next $\Delta t$ state embeddings from the lastly visited state embedding in the demonstration. It worth to note that the last visited state embedding must be included in the part of policy input. Thus, we should only compare the agent's current state embedding with the state embeddings in the demonstration segment provided into the policy for imitation reward. Therefore, the proper value of $\Delta t$ should be less than $m = 10$ and we consider $\Delta t = 2, 4, 8$.

On the simple environments such as Apple-Gold and Deep Sea, we could see in Figure 20a and 20b the different values of $\Delta t$ do not influence the performance a lot.

As shown in Figure 20c and 20d, on more challenging environments where the demonstration trajectory is longer and it's harder to learn to imitate the demonstration, it's better to set a larger value of $\Delta t$ and we can provide imitation reward to the agent and encourage the imitation learning more generously. In general, allowing the agent some flexibility of imitation by setting $\Delta t$ close to $m$ works well.

In summary, $\Delta t = 8$ is a proper value for all of our primary experiment environments.

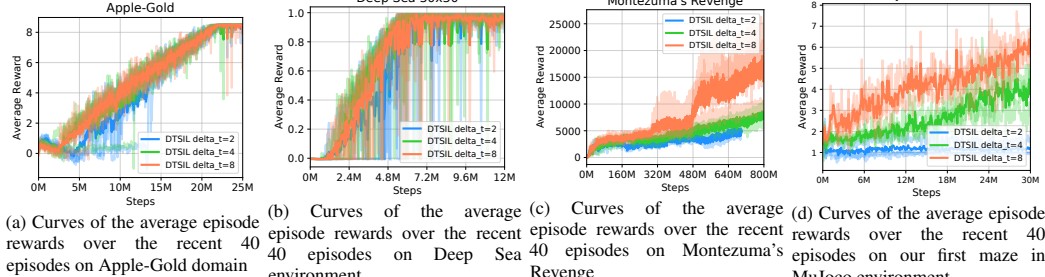

(a) Curves of the average episode rewards over the recent 40 episodes on Apple-Gold domain

(b) Curves of the average episode rewards over the recent 40 episodes on Deep Sea environment

(c) Curves of the average episode rewards over the recent 40 episodes on Montezuma's Revenge

(d) Curves of the average episode rewards over the recent 40 episodes on our first maze in MuJoco environment

Figure 20: Learning curves averaged over at least 3 independent runs. The curves in dark colors are the average of the curves in light colors.

# K    STUDY OF THE STOCHASTICITY IN THE ENVIRONMENT

As listed in Table 3, we consider stochastic environments for all the primary experiments, including the Apple-Gold domain, Montezuma's Revenge, Pitfall, and Mujoco maze. We set the initial state of the agent to be random. On the Apple-Gold domain, the agent takes 3 random steps in the left bottom part of the maze before the episode starts. On Montezuma's Revenge, the mechanism of random initial no-ops is one of the standard ways to introduce stochasticity in the environment (Machado et al., 2017), as in previous work (Mnih et al., 2015).

To show the difficulty in policy learning introduced by the stochastic environments, we tried to memorize and repeat the action sequence in the demonstration trajectory and check whether the agent could successfully visit the state of interest by following the demonstration. In Figure 21c and 22c, it is clear that the success ratio in imitation is much lower than DTSIL, because the agent could not successfully follow the demonstration by just repeating the action sequence step by step. Especially on Montezuma's Revenge, with random initial no-ops, the state of the enemy and electricity beam when the agent starts moving is randomized. Thus it could not successfully avoid death by just repeating the stored action sequence.

Obviously, when the environment is completely deterministic, repeating the action sequence could perfectly guide the agent to the state of interest. However, with the stochasticity of the random initial state, just memorizing the action sequence is sufficient to make the agent revisit novel regions. Thus, the agent could not revisit the novel regions as efficiently as DTSIL to discover better trajectories and converge to a better total episode reward, as shown in Figure 21a, 21b,22a, 22b .

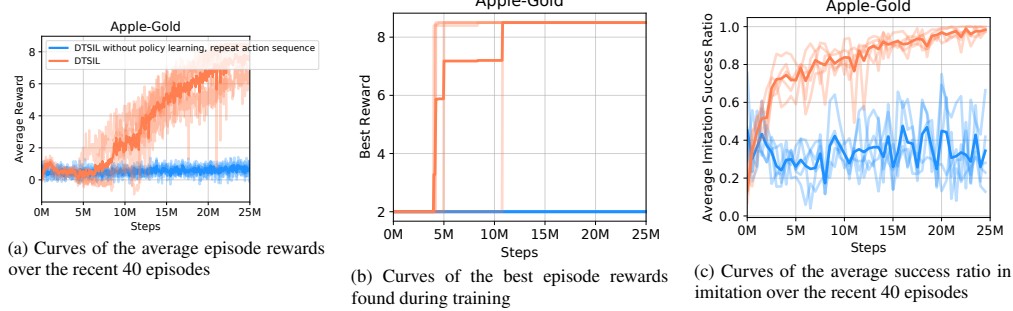

(a) Curves of the average episode rewards over the recent 40 episodes

(b) Curves of the best episode rewards found during training

(c) Curves of the average success ratio in imitation over the recent 40 episodes

Figure 21: Learning curves averaged over 5 independent runs. The curves in dark colors are the average of the curves in light colors. The Apple-Gold domain is with random initial state of the agent.

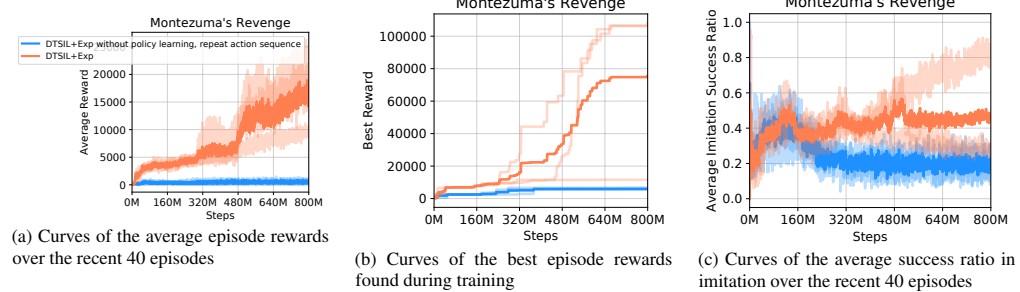

(a) Curves of the average episode rewards over the recent 40 episodes

(b) Curves of the best episode rewards found during training

(c) Curves of the average success ratio in imitation over the recent 40 episodes

Figure 22: Learning curves averaged over 3 independent runs. The curves in dark colors are the average of the curves in light colors. The Montezuma's Revenge is with random initial delay of the agent.

## L    RANDOM EXPLORATION WITH EPSILON-GREEDY POLICY

In this section, we investigate an additional baseline method, random exploration with epsilon-greedy policy, which is a traditional exploration method in reinforcement learning problems. We consider combining epsilon-greedy policy with PPO or DQN framework and run experiments on Apple-Gold, Deep Sea, and Montezuma's Revenge. Our implementation is based on OpenAI baselines (Dhariwal et al., 2017).

As shown in Figure 23, 24, 25, we searched different values of the hyper-parameters for the scheduling of the epsilon, though the performance is not better than DTSIL. Especially, random exploration performs poorly on Montezuma's Revenge. The average score is less than 100, which is consistent with the experimental results from previous works (Mnih et al., 2015; Schulman et al., 2017).

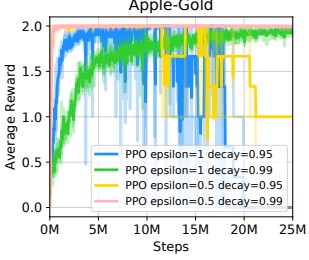 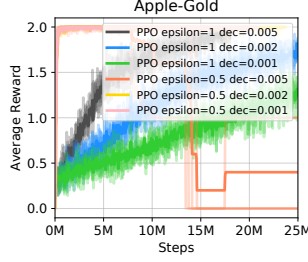 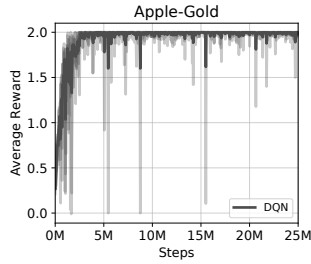

(a) With PPO framework, the value of $\epsilon$ is scheduled in a exponential way (i.e. $\epsilon \leftarrow \epsilon \times$ decay). . The value of $\epsilon$ starts from epsilon as annotated in the legends and decreases until 0.

(b) With PPO framework, the value of $\epsilon$ is scheduled in a linear way (i.e. $\epsilon \leftarrow \epsilon -$ decrement). The value of $\epsilon$ starts from epsilon as annotated in the legends and decreases until 0.

(c) With DQN framework, the value of $\epsilon$ is scheduled in the linear way, from 1 (at the start of training) to 0 (at the end of training)

Figure 23: Learning curves of the average episode rewards over the recent 40 episodes. On the Apple-Gold domain, DTSIL reaches episode reward of 8.5 as shown in Section 4.1 while the random exploration with epsilon-greedy policy gets stuck at the episode reward of 2.

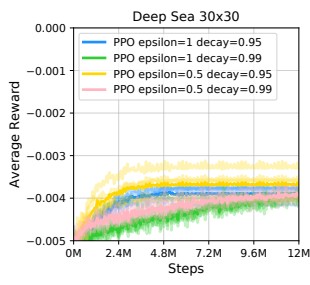 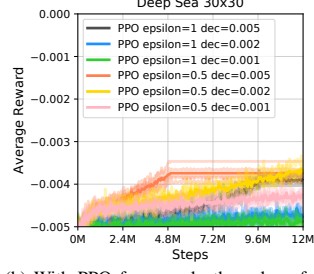 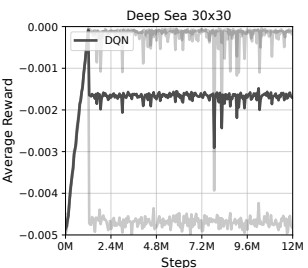

(a) With PPO framework, the value of $\epsilon$ is scheduled in a exponential way (i.e. $\epsilon \leftarrow \epsilon \times$ decay). The value of $\epsilon$ starts from epsilon as annotated in the legends and decreases until 0.

(b) With PPO framework, the value of $\epsilon$ is scheduled in a linear way (i.e. $\epsilon \leftarrow \epsilon -$ decrement). The value of $\epsilon$ starts from epsilon as annotated in the legends and decreases until 0.

(c) With DQN framework, the value of $\epsilon$ is scheduled in the linear way, from 1 (at the start of training) to 0 (at the end of training)

Figure 24: Learning curves of the average episode rewards over the recent 40 episodes. On the Deap Sea environment with $N = 30$, DTSIL reaches episode reward of 1 as shown in Appendix C.3 while the random exploration with epsilon-greedy policy gets stuck at the negative episode reward.

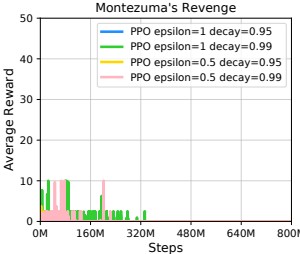

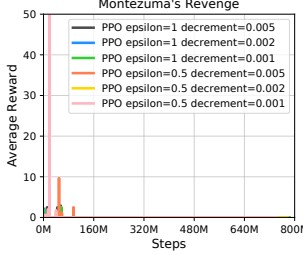

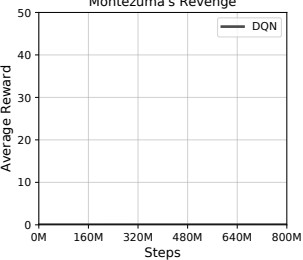

(a) With PPO framework, the value of $\epsilon$ is scheduled in a exponential way (i.e. $\epsilon \leftarrow \epsilon \times decay$). The value of $\epsilon$ starts from epsilon as annotated in the legends and decreases until 0.

(b) With PPO framework, the value of $\epsilon$ is scheduled in a linear way (i.e. $\epsilon \leftarrow \epsilon - decrement$). The value of $\epsilon$ starts from epsilon as annotated in the legends and decreases until 0.

(c) With DQN framework, the value of $\epsilon$ is scheduled in the linear way, from 1 (at the start of training) to 0 (at the end of training)

Figure 25: Learning curves of the average episode rewards over the recent 40 episodes. On the Montezuma's Revenge, DTSIL reaches episode reward over 20,000 as shown in Section 4.2 while the random exploration with epsilon-greedy policy could not reaches score over 100.

