# OpenReview forum: "Self-Imitation Learning via Trajectory-Conditioned Policy for Hard-Exploration Tasks"
_ICLR.cc/2020/Conference — Reject_

### Official Review · AnonReviewer2 · 2019-10-23
**Official Blind Review #2**

**Rating:** 1

**Review:**

The paper addresses the challenge of hard exploration tasks. The approach taken is to apply self-imitation to a diverse selection of trajectories from past experience -- practice re-doing the strangest things you've ever done. This is claimed to drive more efficient exploration in sparse-reward problems, leading to SOTA results for Montezuma's Revenge without certain common aides.

The approach is incompletely motivated. Why trajectory-conditioned policy over just goal-conditioned policy? The note in the related work section doesn't paint a clear enough picture. The trajectory buffer management strategy feels complex. Why this use strategy specifically? Could a simpler design be ruled out? In 2019 (post Go-Explore), it's not clear Montezuma's revenge poses a significant exploration challenge -- exploration doesn't even need to be interleaved with learning. Why are these three the right domains to show off these techniques?

This reviewer moves to reject the paper primarily for not balancing the high complexity of the solution to the lower difficulty of the problem. Pure-exploration algorithms (Go-Explore), not burdened by interleaving policy learning, achieve far superior scores. If the authors want to escape the shadow of this kind of technique which cheats by some framings of RL, more appropriate demonstration environments must be selected.

**Experience Assessment:**

I have published one or two papers in this area.

**Review Assessment: Checking Correctness Of Derivations And Theory:**

N/A

**Review Assessment: Checking Correctness Of Experiments:**

I assessed the sensibility of the experiments.

**Review Assessment: Thoroughness In Paper Reading:**

I read the paper at least twice and used my best judgement in assessing the paper.

---

> ### Author Response · Authors · 2019-11-15
> **Response to Review #2 (Part 2/2)**
>
> Second, it is true that “Pure-exploration algorithms (Go-Explore), not burdened by interleaving policy learning, achieve far superior scores” on Montezuma’s Revenge with the direct state-reset function and deterministic environments. However, on Montezuma’s Revenge without such strong assumptions, which has been well-known as a hard-exploration game for a long time and has been studied by many previous works (Ostrovski et al., 2017; Tang et al., 2017; Burda et al., 2018; Pohlen et al., 2018), to our best knowledge, our work is the first to successfully train the agent to consistently proceed to the second floor and achieve a score over 20,000 without the help of human expert demonstrations. Perhaps our method is not quite simple, but there is no simpler method yet for Montezuma’s Revenge (a notorious sparse-reward problem with moderate stochasticity) without relying on any of state-reset function or human expert demonstrations that can achieve a competitive score to ours.
>
> In conclusion, we are studying a more difficult problem than Go-Explore. For this problem, no previous work (including Go-Explore, as discussed in Appendix H) could perform better than our method. Therefore, we believe the existence of Go-Explore should not be the ground for a rejection or undervaluing our work.
>
> Please note that our paper does not aim solely at solving Montezuma’s Revenge. We would like to study hard-exploration tasks with sparse and misleading rewards. We selected Montezuma’s Revenge, which is a notoriously difficult hard-exploration environment in the literature, as one instance. To fully support our conclusion, we also conducted environments on other interesting domains, including Apple-Gold, Deep Sea and MuJoco maze. We expect that our method could work for real-world tasks such as robotic manipulation, but we leave as future work.
>
> We would like to ask the reviewer to reconsider the significance and difficulty of the problems we are studying. We strongly believe that the hard-exploration problem (such as Montezuma’s Revenge) without a state-reset function, without human expert demonstration and with stochasticity (both in terms of initial state and consequences of taken actions) is a very difficult problem. We are not aware of any publications approaching it with comparable success. We appreciate your time and reconsideration.
>
> References:
> Kulkarni, T. D., Narasimhan, K., Saeedi, A., & Tenenbaum, J. (2016). Hierarchical deep reinforcement learning: Integrating temporal abstraction and intrinsic motivation. In Advances in neural information processing systems (pp. 3675-3683).
> Liu, E. Z., Keramati, R., Seshadri, S., Guu, K., Pasupat, P., Brunskill, E., & Liang, P. (2018). Learning Abstract Models for Long-Horizon Exploration.
> Ostrovski, G., Bellemare, M. G., van den Oord, A., & Munos, R. (2017, August). Count-based exploration with neural density models. In Proceedings of the 34th International Conference on Machine Learning-Volume 70 (pp. 2721-2730). JMLR. org.
> Tang, H., Houthooft, R., Foote, D., Stooke, A., Chen, O. X., Duan, Y., ... & Abbeel, P. (2017). # Exploration: A study of count-based exploration for deep reinforcement learning. In Advances in neural information processing systems (pp. 2753-2762).
> Burda, Y., Edwards, H., Storkey, A., & Klimov, O. (2018). Exploration by random network distillation. arXiv preprint arXiv:1810.12894.
> Pohlen, T., Piot, B., Hester, T., Azar, M. G., Horgan, D., Budden, D., ... & Hessel, M. (2018). Observe and look further: Achieving consistent performance on atari. arXiv preprint arXiv:1805.11593.

---

> > ### Comment · AnonReviewer2 · 2019-11-15
> > **Significance needs to be demonstrated rather than suggested**
> >
> > This reviewer agrees with the authors on the significance of the challenge of hard-exploration problems in stochastic where neither state-reset functionality nor human demonstrations are available. Please do keep working in this area and continue to recruit others to work on this challenge.
> >
> > We need this field to move beyond task formulations that admit unsatisfactory-feeling solutions (Go-Explore). They way to get there is not to ignore or avoid using the exploits they used, but to shift our attention to tasks where those exploits no longer work. After Go-Explore, work that attempts to address the exploration challenge needs to somehow get in contact with a challenge that this algorithm can't address. Not all experiments need to be run in the extra-challenging domain, but at least some should to ensure we are addressing the real problem rather than just what remains after the initial exploit pathways have been removed.

---

> ### Author Response · Authors · 2019-11-15
> **Response to Review #2 (Part 1/2)**
>
> Dear Reviewer #2:
>
> Thank you for the comments.
>
> >>> Why trajectory-conditioned policy over just goal-conditioned policy? The note in the related work section doesn't paint a clear enough picture.
>
> The trajectory-conditioned policy can be thought of as an instance of the goal-conditioned policy, though our “goal” is the trajectory with rich intermediate information about how to achieve the final goal state, thus making it easier for the agent to reach the goal.
>
> Let’s consider other formulations of goal-conditioned policy. If the goal is only the single final state (Kulkarni et al., 2016), it may be difficult to visit the goal state far away from the initial state, especially when the goal is only visited by the agent for just a few times. A good example is a game like Montezuma’s Revenge or Pitfall where there could be many dangers and obstructions along the way to the goal state (e.g., thousands of steps away from the initial state). If the goal includes intermediate information (e.g., a sequence of a small number of sub-goals) aiding the agent towards the final goal state (Liu et al., 2018), the problem becomes easier but it may be still nontrivial to reach the individual sub-goals for long-horizon problems. In addition, learning such a goal-conditioned policy may still require substantial amounts of samples. Our trajectory-conditioned policy is one instance of including the intermediate but more dense information in the goal, making it easier to imitate the previous trajectory with dense imitation reward, and we empirically show that it works well on various domains. We agree there could be alternative (potentially simpler) design choices for our trajectory-conditioned policy, which we will explore in future work.
>
> >>> This reviewer moves to reject the paper primarily for not balancing the high complexity of the solution to the lower difficulty of the problem. Pure-exploration algorithms (Go-Explore), not burdened by interleaving policy learning, achieve far superior scores.
>
> We respectfully disagree that the problems we investigated in this paper, especially Montezuma’s Revenge and Pitfall, are of “lower difficulty” due to the existence of Go-Explore.
>
> First, as we mentioned in Related Work, it is worth noting that the success of Go-Explore heavily relies on the assumption that the environment can be reset to an arbitrary state and the environment is completely deterministic in the exploration phase. We argue that this assumption is infeasible in real-life environments where a high-fidelity simulator may not be available (such as complex robotic tasks) and takes an unfair advantage over the “reset-free” methods. When there is no direct state-reset function and there is stochasticity in the environment, memorizing the past action sequence will not lead the agent to the state of interest (we added these experiments in Appendix K). Therefore in this setting, a more sophisticated “policy learning” is necessary to enable revisiting states of interest. One contribution of our paper is to remove the reliance on the assumption by learning a trajectory-conditioned policy for visiting diverse regions. Therefore, our method could work well in environments without a simulator.

---

### Official Review · AnonReviewer3 · 2019-10-24
**Official Blind Review #3**

**Rating:** 3

**Review:**

The authors identify and address the problem of sub-optimal and myopic behaviors of self-imitation learning in environments with sparse rewards. The authors propose DTSIL to learn a trajectory-conditioned policy to imitate diverse trajectories from the agent’s own past experience. Unlike other self-imitation learning methods, the proposed method not only leverages sub-trajectories with high rewards, but lower-reward trajectories to encourage agent exploration diversity. The authors claim the proposed method to be more likely to find a global optimal solution.

Overall, this paper is well-written with comprehensive experimental results. The proposed trajectory-conditioned policy sounds, since rewarded trajectory carries significant information of the goal in the exploration problem. Extensive experimental results demonstrated the effectiveness of the proposed DTSIL. However, I have a few concerns below, that prevent me from giving a direct acceptance.

1. The proposed DTSIL changes the original MDP with sparse reward to an MDP with denser reward, which allows the training process to explore more in the “space” closer to the collected high reward trajectories. Such “exploration” sounds promising. However, it would be nice to compare it with traditional reinforcement learning (e.g., with \epsilon-greedy policy for random exploration)?

2. In appendix D, the authors discussed what the parameter \delta_t controls, however, it is unclear how \delta_t should be chosen in implementation. The authors did not explain how \delta_t was selected in their experiments. Choosing the right \delta_t may be hard, but it would be nice to introduce what “heuristics” the authors used and suggest to readers.


**Experience Assessment:**

I have published in this field for several years.

**Review Assessment: Checking Correctness Of Derivations And Theory:**

N/A

**Review Assessment: Checking Correctness Of Experiments:**

I assessed the sensibility of the experiments.

**Review Assessment: Thoroughness In Paper Reading:**

I read the paper thoroughly.

---

> ### Author Response · Authors · 2019-11-15
> **Response to Review #3**
>
> Dear Reviewer #3:
>
> Thank you for the clear and constructive feedback.
>
> For question 1, as shown in Appendix L, we conducted the experiments with the traditional exploration mechanism of the epsilon-greedy strategy. We combined epsilon-greedy policy with PPO (Schulman et al. 2017) framework and DQN (Mnih et al. 2015) framework and searched many values of the hyper-parameters for the epsilon scheduling. Even though we have put much effort to push the experiments with random exploration, the performance is much worse than DTSIL on these hard-exploration tasks. Especially for Montezuma’s Revenge, the epsilon-greedy policy with random exploration achieves a score less than 100, which is consistent with the experimental results from previous works (Mnih et al. 2015, Schulman et al. 2017).
>
> Also, we would like to emphasize that our main baseline method, i.e. count-based exploration, is also one of the classic, well-performing exploration mechanisms (Strehl & Littman 2005, Kolter & Ng 2009). However, we found that a simple combination of count-based exploration with standard RL method (e.g., PPO) still requires encountering many similar high-reward episodes to learn good behavior and empirically performs worse than our proposed method. We believe that it is because our method can better leverage a few samples of high-reward trajectories by learning to explore the variants of those past trajectories.
>
> For question 2, we added Appendix J for the ablation study of the hyper-parameter $\Delta t$. The only constraint is that $\Delta t$ should be less than m (length of demonstration segment to imitate as input into the policy; we set m=10 for all our experiments by considering computational constraints). In general, we found that allowing the agent some flexibility of imitation by setting $\Delta t$ close to m works well. For easy domains, $\Delta t=2,4,8$ does not show much difference in policy performance. For more difficult domains, $\Delta t=8$ works better because we provide imitation rewards more leniently to the agent to encourage imitation of the demonstration. In summary, $\Delta t=8$ performs well for all of our primary experiment environments.
>
> We hope that our response above will address your concerns and thank you again for the suggestions.
>
> References:
> Mnih, V., Kavukcuoglu, K., Silver, D., Rusu, A. A., Veness, J., Bellemare, M. G., ... & Petersen, S. (2015). Human-level control through deep reinforcement learning. Nature, 518(7540), 529.
> Schulman, J., Wolski, F., Dhariwal, P., Radford, A., & Klimov, O. (2017). Proximal policy optimization algorithms. arXiv preprint arXiv:1707.06347.
> Strehl, A. L., & Littman, M. L. (2005, August). A theoretical analysis of model-based interval estimation. In Proceedings of the 22nd international conference on Machine learning (pp. 856-863). ACM.
> Kolter, J. Z., & Ng, A. Y. (2009, June). Near-Bayesian exploration in polynomial time. In Proceedings of the 26th Annual International Conference on Machine Learning (pp. 513-520). ACM.

---

### Official Review · AnonReviewer1 · 2019-10-25
**Official Blind Review #1**

**Rating:** 6

**Review:**

Note: the style-formatting of this paper has been heavily tweaked, and so the evaluation should be calibrated for a 9-page paper.

This paper proposes an approach for diverse self-imitation for hard exploration problems.  The idea is leverage recently proposed self-imitation approaches for learning to imitate good trajectories generated by the policy itself.  By encouraging diversity in the pool of trajectories for self-imitation, the idea is to encourage faster learner -- this basic concept is also used in approaches like prioritized experience replay, albeit at the entire trajectory level rather than individual state/action level.

The authors view this approach as a generalization of Go-Explore, since it does not rely on having a reset mechanism.  However, I think this discussion has a lot of subtle nuances pertaining to the stochasticity of the environment (which the authors acknowledge).  For instance, if the environment is deterministic, then why not just do something like Go-Explore, since state-reset is just memorizing a deterministic action sequence?

The empirical results are very strong, achieving state-of-the-art results for any approach not reliant on a reset mechanism.  All the primary experiments appear to be for deterministic environments.  The results on stochastic environments (in the Appendix) seem pretty weak (but please correct me if I'm mistaken here).  So one major question is whether Go-Explore is a scientifically appropriate benchmark to compare with for this setting.

In summary, I'm willing to be convinced that this is an interesting and scientifically novel result.  I have some concerns as expressed above.


**** After Author Response ****
Thanks for the response.  I'm willing to raise my score to weak accept.

I think the authors did a reasonable job addressing my specific questions.  Some further reflection revealed to me that there is a huge opportunity to scientifically investigate how stochasticity impacts the proposed algorithm.  For instance, one could conduct a systematic study (say of the Apple domain) where one varies the degree of stochasticity and measures how the performance the proposed algorithm changes, perhaps relative to Go-Explore on the purely deterministic version of the environment.  It seems a bit of a cop-out to say that Go-Explore is not applicable, and misses out a huge opportunity for real scientific understanding.

**Experience Assessment:**

I have read many papers in this area.

**Review Assessment: Checking Correctness Of Derivations And Theory:**

I carefully checked the derivations and theory.

**Review Assessment: Checking Correctness Of Experiments:**

I carefully checked the experiments.

**Review Assessment: Thoroughness In Paper Reading:**

I read the paper thoroughly.

---

> ### Author Response · Authors · 2019-11-15
> **Response to Review #1**
>
> Dear Reviewer #1:
>
> Thanks for your detailed and helpful feedback. We have simply moved the implementation details in Section 4 to the Appendix so that the entire paper can fit into 8 pages without much tweaking of style-format.
>
> About the environment setting, as explained at the start of section 4.2, 4.3 and listed in Table 3, Appendix E, we considered the stochastic environments for all the primary experiments, including Apple-Gold, Montezuma’s Revenge, Pitfall, and MuJoco. The initial state of the agent in the experiment is randomized. The mechanism of initial random no-ops is one of the standard ways to introduce stochasticity in the Atari environment (Machado et al., 2018). The mechanism of random initial location from a Gaussian distribution in MuJoco maze is the same as in standard MuJuco tasks (Brockman et al.,2016).
>
> In Appendix C.1, we showed DTSIL outperforms the baselines and achieves near-optimal episode reward with different forms of stochasticity in the Apple-Gold domain (i.e. random initial location of the agent, sticky action, and random initial location of the treasure). In the Apple-Gold domain, as we introduced in Figure 1, the agent achieves reward +1 when collecting an apple, gets reward +10 when collecting the treasure, but gets reward -0.05 when taking a step in the rocky region. Therefore, with the time limit of 45 steps, the optimal trajectory is to go through the rocky region for 30 steps and reach the treasure to get the total episode reward of 8.5. Different from the baselines, DTSIL successfully finds the optimal path and converges to good behavior.
>
> In order to show the difficulty in policy learning in these stochastic environments, we added an additional baseline for DTSIL in Appendix K. Specifically, we stored and repeated the action sequence from the demonstration trajectory. When the environment is deterministic, repeating the action sequence should perfectly lead the agent to imitate the demonstration and reach the final state of interest. However, as shown in Figure 21 and 22 in Appendix K, on environments with random initial states (which is a standard type of moderate-degree stochasticity in the literature), memorizing the action sequence is not sufficient. The success ratio in imitation is much lower than DTSIL. Thus the agent could not revisit the novel regions as efficiently as DTSIL to discover better trajectories and converge to a better total episode reward.
>
> We hope this information will address your concerns about the deterministic environments and thank you again for your comments.
>
> References:
> Machado, M. C., Bellemare, M. G., Talvitie, E., Veness, J., Hausknecht, M., & Bowling, M. (2018). Revisiting the arcade learning environment: Evaluation protocols and open problems for general agents. Journal of Artificial Intelligence Research, 61, 523-562.
> Brockman, G., Cheung, V., Pettersson, L., Schneider, J., Schulman, J., Tang, J., & Zaremba, W. (2016). Openai gym. arXiv preprint arXiv:1606.01540.

---

### Decision · Program_Chairs · 2019-12-19

**Decision:**

Reject

**Comment:**

This paper addresses the problem of exploration in challenging RL environments using self-imitation learning. The idea behind the proposed approach is for the agent to imitate a diverse set of its own past trajectories. To achieve this, the authors introduce a policy conditioned on trajectories. The proposed approach is evaluated on various domains including Atari Montezuma's Revenge and MuJoCo.

Given that the evaluation is purely empirical, the major concern is in the design of experiments. The amount of stochasticity induced by the random initial state alone does not lead to convincing results regarding the performance of the proposed approach compared with baselines (e.g. Go-Explore). With such simple stochasticity, it is not clear why one could not use a model to recover from it and then rely on an existing technique like Go-Explore. Although this paper tackles an important problem (hard-exploration RL tasks), all reviewers agreed that this limitation is crucial and I therefore recommend to reject this paper.